# Impression learning: Online representation learning with synaptic plasticity

**Colin Bredenberg**
Center for Neural Science
New York University
cjb617@nyu.edu

**Benjamin S. H. Lyo**
Center for Neural Science
New York University
blyo@nyu.edu

**Eero P. Simoncelli**
Center for Neural Science,
New York University
Flatiron Institute, Simons Foundation
eero.simoncelli@nyu.edu

**Cristina Savin**
Center for Neural Science,
Center for Data Science
New York University
csavin@nyu.edu

## Abstract

Understanding how the brain constructs statistical models of the sensory world remains a longstanding challenge for computational neuroscience. Here, we derive an unsupervised local synaptic plasticity rule that trains neural circuits to infer latent structure from sensory stimuli via a novel loss function for approximate online Bayesian inference. The learning algorithm is driven by a local error signal computed between two factors that jointly contribute to neural activity: stimulus drive and internal predictions — the network's 'impression' of the stimulus. Physiologically, we associate these two components with the basal and apical dendrites of pyramidal neurons, respectively. We show that learning can be implemented online, is capable of capturing temporal dependencies in continuous input streams, and generalizes to hierarchical architectures. Furthermore, we demonstrate both analytically and empirically that the algorithm is more data-efficient than a three-factor plasticity alternative, enabling it to learn statistics of high-dimensional, naturalistic inputs. Overall, the model provides a bridge from mechanistic accounts of synaptic plasticity to algorithmic descriptions of unsupervised probabilistic learning and inference.

## 1 Introduction

Sensory systems are faced with a task analogous to the scientific process itself: given a steady stream of raw data, they must extract meaningful information about its underlying structure. Because the true underlying structure of the data is rarely accessible, this "representation learning" must be largely unsupervised. Framing perception in the language of Bayesian inference has proven fruitful in perceptual and cognitive science [1–4], but has been difficult to connect to biology, because we still lack a satisfactory account of how the machinery of Bayesian inference and learning is implemented in neural circuits [5, 6].

Past work includes several examples of circuits that simultaneously learn a top-down generative model of incoming stimuli and perform approximate inference with respect to these models. These differ in the nature of the approximation, from maximum a posteriori estimation [7], to efficient population codes that embed prior structure [8] to either parametric [9, 10] or sampling-based [11] variational inference. Learning generally takes the form of optimizing a probabilistic objective, either by backpropagation [9, 10] or through local parameter updates, which match biological learning more

35th Conference on Neural Information Processing Systems (NeurIPS 2021).

closely [11, 7, 12, 13]. While these models are mostly restricted to static stimuli, several instances also operate over time [14–16].

Developing biologically plausible learning rules that are applicable to temporally-structured data is hampered by the fact that optimizing a probabilistic objective function in such contexts requires access to non-local information across space and time. Previous research on local approximations to credit assignment in BP address spatial credit assignment by ascribing differential functions to the apical and basal dendrites of pyramidal neurons in cortex, where apical dendrites are hypothesized to receive top-down learning signals, and basal dendrites receive bottom-up sensory signals [17–25]. Locally implementing temporal credit assignment is a bigger challenge [26, 27].

Our work, which we have dubbed 'impression learning' (IL), combines the tradition of probabilistic learning [11, 14] with these recent developments in local optimization, in order to learn dynamic stimuli concurrently with perception. We propose a network architecture in which top-down stimulus predictions arriving at the apical dendrites of neurons influence both network dynamics and synaptic plasticity, allowing the network to concurrently learn a probabilistic model of the stimuli and an approximate inference computation. We provide a mathematical derivation of synaptic plasticity rules that approximate gradient descent on a novel unsupervised loss function, along with detailed analyses of the biases induced by this approximation. We explore the empirical and mathematical relationships between IL and three other methods: backpropagation (BP) [28], the Wake-Sleep (WS) algorithm [29], and a specific form of neural variational inference (NVI*) [30, 31]. We further demonstrate that IL scales to naturalistic stimuli and multilayer network architectures [1].

## 2 Probabilistic inference and local learning in a recurrent circuit

We construct a network of neurons that aims to learn a generative model of the temporal sequence of stimuli that it receives, $p_m(\mathbf{r}, \mathbf{s}) = \prod_{t=0}^{T} p_m(\mathbf{r}_t, \mathbf{s}_t | \mathbf{r}_{t-1})$, in which $\mathbf{s}$ represents stimuli in an input layer.[2] The latent variables $\mathbf{r}$ are not defined by a physical model of the stimulus environment, but are learned in an unsupervised manner to provide the best generative explanation of stimuli received. We assume that stimuli are generated by a *true* probability distribution $p(\mathbf{s}|\mathbf{z})$, where $\mathbf{s}$ corresponds to the first layer of neural activations in an early sensory layer, and vector $\mathbf{z} \sim p(\mathbf{z})$ corresponds to the environmental factors which jointly caused that activity. Because learning is unsupervised, we do not enforce explicit correspondence between the internal and true latent features, $\mathbf{r}$ and $\mathbf{z}$, only a correspondence between model predictions and ground truth stimuli. We also assume that the network performs online inference with respect to its model, inferring the corresponding latent cause $\mathbf{r}$ using Bayes' rule: $p_m(\mathbf{r}|\mathbf{s}) = p_m(\mathbf{r}, \mathbf{s})/p_m(\mathbf{s})$. Because the network won't, in general, be able to explicitly calculate Bayes' rule, we will assume that the network learns an *approximate* inference distribution $q(\mathbf{r}|\mathbf{s})$, which it attempts to bring 'close' to $p_m(\mathbf{r}|\mathbf{s})$. This joint process of learning and inference, known as Bayesian latent feature extraction, provides a general framework for conceptualizing early sensory processing in the brain [5]. In subsequent sections, we will write a loss function for this general latent feature extraction objective, and show how local modifications at apical and basal synapses can perform approximate gradient descent on this loss.

**Loss function**   The loss function that we propose will produce a learning algorithm where neurons alternate between sampling from the model, $p_m$, and performing approximate inference according to $q$ in response to real stimuli received from $p(\mathbf{s}|\mathbf{z})$. This alternation will allow the network to learn online in a way that minimally perturbs the continuity of perception. First, consider two families of hybrid probability distributions, which we denote in shorthand $\tilde{q}_\theta$ and $\tilde{p}_\theta$:

$$\tilde{q}_\theta = \prod_{t=0}^{T} \tilde{q}_t(\mathbf{r}_t, \mathbf{s}_t | \mathbf{z}_t, \lambda_t; \theta) = \prod_{t=0}^{T} \left( q(\mathbf{r}_t|\mathbf{s}_t; \theta_q) p(\mathbf{s}_t|\mathbf{z}_t) \right)^{\lambda_t} p_m(\mathbf{r}_t, \mathbf{s}_t|\mathbf{r}_{t-1}, \lambda_t; \theta_p)^{1-\lambda_t}$$

$$\tilde{p}_\theta = \prod_{t=0}^{T} \tilde{p}_t(\mathbf{r}_t, \mathbf{s}_t | \mathbf{z}_t, \lambda_t; \theta) = \prod_{t=0}^{T} \left( q(\mathbf{r}_t|\mathbf{s}_t; \theta_q) p(\mathbf{s}_t|\mathbf{z}_t) \right)^{1-\lambda_t} p_m(\mathbf{r}_t, \mathbf{s}_t|\mathbf{r}_{t-1}, \lambda_t; \theta_p)^{\lambda_t}, \quad (1)$$

where a collection of binary random variables $\lambda_t$ determines whether, at a given time step, sampling occurs due to $q(\mathbf{r}_t|\mathbf{s}_t; \theta_q) p(\mathbf{s}_t|\mathbf{z}_t)$ or $p_m(\mathbf{r}_t, \mathbf{s}_t|\mathbf{r}_{t-1}, \lambda_t; \theta_p)$, and the full parameter space is denoted

---

[1]Code provided at: `https://github.com/colinbredenberg/Impression-Learning-Camera-Ready`.
[2]We use the shorthand notation '$\mathbf{s}$' to refer to the $N \times T$ matrix of stimuli across time.

$\theta = [\theta_p, \theta_q]$. We define an objective of the form:

$$\mathcal{L} = \mathbb{E}_{\lambda, \mathbf{z}} \left[ KL[\tilde{q}_\theta || \tilde{p}_\theta] \right]$$
$$= \mathbb{E}_{\lambda, \mathbf{z}} \left[ \int [\log \tilde{q}_\theta - \log \tilde{p}_\theta] \, \tilde{q}_\theta \, d\mathbf{r} d\mathbf{s} \right]. \tag{2}$$

This loss provides a generalization of the widely-used evidence lower bound (ELBO), which corresponds to the case $\lambda_t = 1 \; \forall t$. Importantly, we can show that $\mathcal{L} = 0$ if and only if $q(\mathbf{r}_t | \mathbf{s}_t; \theta_q) p(\mathbf{s}_t | \mathbf{z}_t) = p_m(\mathbf{r}_t, \mathbf{s}_t | \mathbf{r}_{t-1}, \lambda_t; \theta_p) \; \forall t$. If this equality were achieved, it would also imply $p_m(\mathbf{r}, \mathbf{s}) = q(\mathbf{r}|\mathbf{s})p(\mathbf{s}|\mathbf{z})$. However, this absolute minimum will not be achievable unless $\mathbf{z}_t$ is deterministic, because $p_m(\mathbf{r}_t, \mathbf{s}_t | \mathbf{r}_{t-1}, \lambda_t; \theta_p)$ has no dependency on the latent variables in the environment. Thus, our goal is inherently unachievable, and different choices of $p(\lambda_t)$ and network architectures may lead to different local minima. However, each choice will incentivize learning a close correspondence between these distributions, and an approximation to gradient descent with respect to *any* choice will lead to local synaptic plasticity rules, making this objective particularly interesting for the computational neuroscience community.

**Update derivation**     We begin by taking the gradient of our new loss w.r.t. $\theta = [\theta_q, \theta_p]$:

$$-\nabla_\theta \mathcal{L} = -\nabla_\theta \mathbb{E}_{\lambda, \mathbf{z}} \left[ \int [\log \tilde{q}_\theta - \log \tilde{p}_\theta] \, \tilde{q}_\theta \, d\mathbf{r} d\mathbf{s} \right]$$
$$= -\mathbb{E}_{\lambda, \mathbf{z}} \left[ \int [\nabla_\theta (\log \tilde{q}_\theta - \log \tilde{p}_\theta)] \, \tilde{q}_\theta \, d\mathbf{r} d\mathbf{s} + \int [\log \tilde{q}_\theta - \log \tilde{p}_\theta] \, \nabla_\theta \tilde{q}_\theta \, d\mathbf{r} d\mathbf{s} \right],$$

where the second equality follows from the product rule. Both integrals are analytically intractable, but if we can write both as expectations, they can be approximated by averaging over samples of $\mathbf{r}$ and $\mathbf{s}$. To accomplish this, we note that $\nabla_\theta \tilde{q}_\theta = \nabla_\theta e^{\log \tilde{q}_\theta} = [\nabla_\theta \log \tilde{q}_\theta] \, \tilde{q}_\theta$, which allows us to rewrite our expression as an expectation over $\mathbf{r}$ and $\mathbf{s}$:

$$-\nabla_\theta \mathcal{L} = -\mathbb{E}_{\lambda, \mathbf{z}} \left[ \int [\nabla_\theta \log \tilde{q}_\theta - \nabla_\theta \log \tilde{p}_\theta] \, \tilde{q}_\theta \, d\mathbf{r} d\mathbf{s} + \int [\log \tilde{q}_\theta - \log \tilde{p}_\theta] \, (\nabla_\theta \log \tilde{q}_\theta) \tilde{q}_\theta \, d\mathbf{r} d\mathbf{s} \right].$$

We also observe that $\int [\nabla_\theta \log \tilde{q}_\theta] \, \tilde{q}_\theta \, d\mathbf{r} d\mathbf{s} = \nabla_\theta \int \tilde{q}_\theta \, d\mathbf{r} d\mathbf{s} = \nabla_\theta 1 = 0$, allowing the elimination of two terms:

$$-\nabla_\theta \mathcal{L} = \mathbb{E}_{\lambda, \mathbf{z}} \left[ \int [\nabla_\theta \log \tilde{p}_\theta] \, \tilde{q}_\theta \, d\mathbf{r} d\mathbf{s} + \int \left[\log \frac{\tilde{p}_\theta}{\tilde{q}_\theta}\right] (\nabla_\theta \log \tilde{q}_\theta) \tilde{q}_\theta \, d\mathbf{r} d\mathbf{s} \right]$$
$$\approx \mathbb{E}_{\lambda, \mathbf{z}} \left[ \int [\nabla_\theta \log \tilde{p}_\theta] \, \tilde{q}_\theta \, d\mathbf{r} d\mathbf{s} + \int \left[\frac{\tilde{p}_\theta}{\tilde{q}_\theta} - 1\right] (\nabla_\theta \log \tilde{q}_\theta) \tilde{q}_\theta \, d\mathbf{r} d\mathbf{s} \right]$$
$$= \mathbb{E}_{\lambda, \mathbf{z}} \left[ \int [\nabla_\theta \log \tilde{p}_\theta] \, \tilde{q}_\theta \, d\mathbf{r} d\mathbf{s} + \int \left[\frac{\tilde{p}_\theta}{\tilde{q}_\theta}\right] (\nabla_\theta \log \tilde{q}_\theta) \tilde{q}_\theta \, d\mathbf{r} d\mathbf{s} \right]$$
$$= \mathbb{E}_{\lambda, \mathbf{z}} \left[ \int [\nabla_\theta \log \tilde{p}_\theta] \, \tilde{q}_\theta \, d\mathbf{r} d\mathbf{s} + \int [\nabla_\theta \log \tilde{q}_\theta] \, \tilde{p}_\theta \, d\mathbf{r} d\mathbf{s} \right]. \tag{3}$$

The approximation in the second line comes from a Taylor expansion of $\log \frac{\tilde{p}_\theta}{\tilde{q}_\theta}$ about 0, i.e. when $\frac{\tilde{p}_\theta}{\tilde{q}_\theta} = 1$ (which introduces a bias to the parameter updates that we examine analytically in Appendix A). This expansion is the core of our derivation, and not all algorithms take this approach: for this reason, in Appendix B and C we show how the properties of our algorithm compare to alternatives (NVI*, BP, or WS).

At this point, we have not yet defined $p(\lambda)$. We'll assume that $\lambda_0 \in \{0, 1\}$, that $p(\lambda_0 = 0) = p(\lambda_0 = 1) = 0.5$, and that the $\lambda$ values alternate deterministically with a 'phase duration' $K$, i.e. $\lambda_{k+1} = 1 - \lambda_k$ if $\mod (k, K) = 0$, and $\lambda_{k+1} = \lambda_k$ otherwise. Under these conditions, the two integrals in Eq. (3) are *equivalent*, and computing our parameter updates only requires sampling from $\tilde{q}$. If we define $\lambda' = 1 - \lambda$, then we have $p(\lambda') = p(\lambda)$ and $\tilde{q}(\mathbf{r}, \mathbf{s}|\mathbf{z}, \lambda; \theta) = \tilde{p}(\mathbf{r}, \mathbf{s}|\mathbf{z}, \lambda'; \theta)$, which

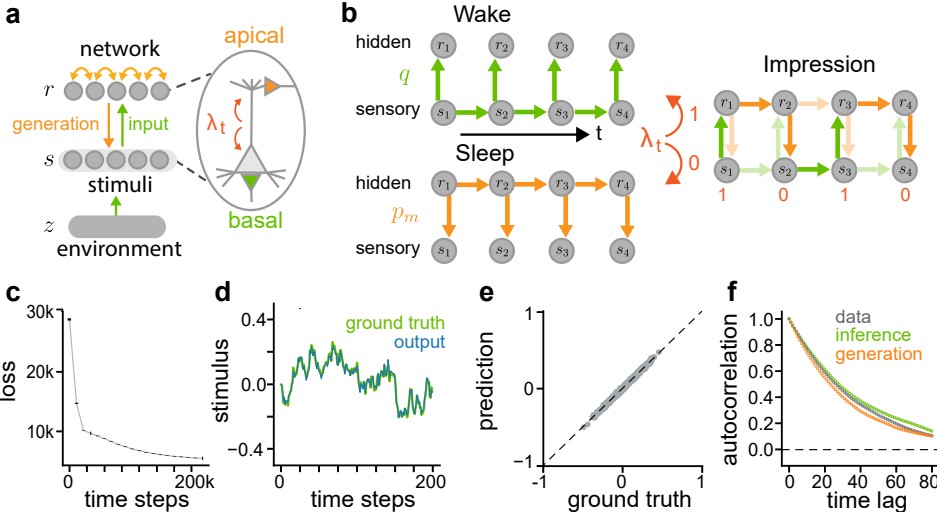

Figure 1: **Network architecture and learning. a.** Model schematic. A neural network receives stimulus inputs at its basal dendrites, and returns lateral and top-down prediction signals via apical synapses. A gate, $\lambda_t$, determines whether apical or basal influences dominate network activity. **b.** Learning schedule: the Wake-Sleep (WS) algorithm (left) trains its synapses by alternating between prolonged periods where $\lambda_t = 1$ (Wake) or $\lambda_t = 0$ (Sleep). In contrast, our IL algorithm alternates rapidly between $\lambda_t = 1$ and $\lambda_t = 0$ with phase duration $K = 2$. **c.** Network loss on the artificial stimulus task. Error bars indicate $\pm 1$ s.e.m. averaged across 20 network realizations. **d.** Comparison between a ground truth stimulus (green) and the network's prediction (blue) for a particular stimulus dimension. **e.** Same comparison across stimulus dimensions. **f.** The autocorrelation function of **r** when the network is performing approximate inference (green; $\lambda_t = 1$), or in generative mode (orange; $\lambda_t = 0$) compared to the autocorrelation of the data (grey).

we make use of as follows:

$$
\begin{aligned}
-\nabla_\theta \mathcal{L} &\approx \mathbb{E}_{\mathbf{z}}\left[\sum_\lambda \left[\int \left[\nabla_\theta \log \tilde{p}_\theta\right] \tilde{q}_\theta \; d\mathbf{r}d\mathbf{s} + \int \left[\nabla_\theta \log \tilde{q}_\theta\right] \tilde{p}_\theta \; d\mathbf{r}d\mathbf{s}\right] \; p(\lambda)\right] \\
&= \mathbb{E}_{\mathbf{z}}\left[\sum_\lambda \int \left[\nabla_\theta \log \tilde{p}_\theta\right] \tilde{q}_\theta \; d\mathbf{r}d\mathbf{s}p(\lambda) + \sum_{\lambda'} \int \left[\nabla_\theta \log \tilde{p}_\theta\right] \tilde{q}_\theta \; d\mathbf{r}d\mathbf{s} \; p(\lambda')\right] \\
&= 2\mathbb{E}_{\mathbf{z}}\left[\sum_\lambda \int \left[\nabla_\theta \log \tilde{p}_\theta\right] \tilde{q}_\theta \; d\mathbf{r}d\mathbf{s} \; p(\lambda)\right].
\end{aligned}
\tag{4}
$$

Using the definitions for $\tilde{q}_\theta$ and $\tilde{p}_\theta$ and the properties of the logarithm gives us the following parameter update rule:

$$
\Delta\theta \propto 2\mathbb{E}_{\lambda_0, \mathbf{z}}\left[\int \left[\sum_t (1 - \lambda_t)\nabla_\theta \log q_t + (\lambda_t)\nabla_\theta \log p_{mt}\right] \tilde{q}_\theta \; d\mathbf{r}d\mathbf{s}\right].
\tag{5}
$$

As we will show below, this parameter update equation produces updates that require only information locally available to synapses, a necessary condition for any biologically-plausible algorithm.

**Basic model**    To make the above general learning procedure concrete, we need to specify how to sample from $\tilde{q}_\theta$, which in turn requires an architecture for performing approximate inference at each time step, $q(\mathbf{r}_t|\mathbf{s}_t; \theta_q)$, and a joint model of stimuli and neural activations, $p_m(\mathbf{r}_t, \mathbf{s}_t|\mathbf{r}_{t-1}, \lambda_t; \theta_p)$. We map these two model components onto neural circuitry, with their own local variables corresponding to **s** and **r**, and segregated synaptic parameters: the 'basal' compartment is dedicated to feedforward inference ($q$, index 'inf') and the 'apical' compartment is dedicated to generative sampling from the model ($p_m$, index 'gen'); this segregation allows their influence on neural dynamics to be selectively gated by $\lambda_t$ (Fig. 1a).

First, the internal generative model of the circuit is implicitly defined by a set of currents to the apical dendritic compartment corresponding to generated samples for the next latent variable, $\mathbf{r}_t^{\text{gen}}$:

$$\mathbf{r}_t^{\text{gen}} = ((1 - k_t)\mathbf{D}_r + k_t\mathbf{I})\,\mathbf{r}_{t-1} + \sigma_r^{\text{gen}}\boldsymbol{\eta}_t \tag{6}$$

$$\mathbf{s}_t^{\text{gen}} = f(\mathbf{D}_s\mathbf{r}_t) + \sigma_s^{\text{gen}}\boldsymbol{\xi}_t, \tag{7}$$

where $\mathbf{D}_r$ is a diagonal transition matrix (constraining generated latent-variables to be independent AR(1) processes), $\mathbf{D}_s$ is a linear decoder, $\mathbf{I}$ is the identity function, $\boldsymbol{\eta}_t, \boldsymbol{\xi}_t \sim \mathcal{N}(0, \mathbf{I})$ are independent white noise samples, and $\sigma_r^{\text{gen}}$ and $\sigma_s^{\text{gen}}$ denote respectively the generative standard deviation for neurons at the stimulus and latent levels. We define $k_t = (1 - \delta(\lambda_t - \lambda_{t-1}))\lambda_t$, with $\delta(\cdot)$ the Dirac delta function; $k_t$ is 1 only if $\lambda_t = 1$ and $\lambda_{t-1} = 0$. We chose a piecewise model (gated by $k_t$) for $\mathbf{r}_t^{\text{gen}}$ because we observed that the statistics of stimuli $\mathbf{s}_t$ given previous activities $\mathbf{r}_{t-1}$ are different if a transition has just occurred ($\lambda_t = 1$ and $\lambda_{t-1} = 0$), which will bias the training of the generative transition parameters $\mathbf{D}_r$. We chose $\mathbf{I}$ for this case, but one could alternatively have a different parametric model for after transitions have occurred. As we will show, adding this condition to our model will never affect the *dynamics* of our network, but will cause learning for $\mathbf{D}_r$ to occur only on time steps when a transition has not just occurred. Nothing in our derivation requires the transition matrix $\mathbf{D}_r$ to be diagonal, but we constrained it in this way to allow for learning independent latent features. As is, $\mathbf{D}_r$ defines the leakiness of the apical dendritic compartment of the neuron; off-diagonal components of the transition matrix would correspond to recurrent synapses. These dynamics define a probability distribution: $p_m(\mathbf{r}, \mathbf{s}) = \prod_{t=0}^{T} p_m(\mathbf{r}_t, \mathbf{s}_t | \mathbf{r}_{t-1}, \lambda_t; \theta_p)$.

Second, we define our inference model, a factorized conditional probability distribution $q(\mathbf{r}|\mathbf{s}) = \prod_{t=0}^{T} q(\mathbf{r}_t|\mathbf{s}_t; \theta_q)$, which applies a feedforward nonlinear transformation to incoming stimuli:

$$\mathbf{r}_t^{\text{inf}} = f(\mathbf{W}\mathbf{s}_t) + \sigma_r^{\text{inf}}\boldsymbol{\eta}_t, \tag{8}$$

where $\mathbf{W}$ denotes the feedforward weights, $\sigma_r^{\text{inf}}$ is the inference standard deviation for neurons at the latent level, and the nonlinearity $f(\cdot)$ is the $\tanh$ function. During inference mode, the stimulus layer receives latent-associated inputs from the environment, further corrupted by the same noise as the internal representation:

$$\mathbf{s}_t^{\text{inf}} = \bar{\mathbf{s}}(\mathbf{z}_t) + \sigma_s^{\text{inf}}\boldsymbol{\xi}_t, \tag{9}$$

where $\sigma_s^{\text{inf}}$ denotes the standard deviation for neurons at the stimulus level, and $\bar{\mathbf{s}}(\mathbf{z}_t)$ is input from external stimuli. During simulations, samples are determined by a combination of $p_m$ and $q$, given by $\tilde{q}_\theta$:

$$\mathbf{r}_t = \lambda_t\mathbf{r}_t^{\text{inf}} + (1 - \lambda_t)\mathbf{r}_t^{\text{gen}} \tag{10}$$

$$\mathbf{s}_t = \lambda_t\mathbf{s}_t^{\text{inf}} + (1 - \lambda_t)\mathbf{s}_t^{\text{gen}}. \tag{11}$$

We interpret these dynamics biologically as network of recurrently connected pyramidal neurons with two sources of input, one to the apical dendrites ($\mathbf{r}_t^{gen}$ or $\mathbf{s}_t^{\text{gen}}$) and one to the basal dendrites ($\mathbf{r}_t^{\text{inf}}$ or $\mathbf{s}_t^{\text{inf}}$). The gating variable $\lambda_t$ determines which input source controls the circuit dynamics.

**Plasticity rule interpretation** Inserting our particular choice of $q_t$ and $p_{mt}$ into our approximate gradient descent derivation, the parameter updates can be interpreted as local synaptic plasticity rules at the basal (for $q_t$) or apical (for $p_{mt}$) compartments of our neuron model:

$$\log q(\mathbf{r}_t|\mathbf{s}_t; \theta_q) = -\frac{1}{2\left(\sigma_r^{\text{inf}}\right)^2}\|\mathbf{r}_t - f(\mathbf{W}\mathbf{s}_t)\|_2^2 + c_q \tag{12}$$

$$\log p_m(\mathbf{r}_t, \mathbf{s}_t|\mathbf{r}_{t-1}, \lambda_t; \theta_p) = -\frac{1}{2(\sigma_r^{\text{gen}})^2}\|\mathbf{r}_t - ((1 - k_t)\mathbf{D}_r + k_t\mathbf{I})\,\mathbf{r}_{t-1}\|_2^2$$

$$-\frac{1}{2(\sigma_s^{\text{gen}})^2}\|\mathbf{s}_t - f(\mathbf{D}_s\mathbf{r}_t)\|_2^2 + c_p, \tag{13}$$

where $c_q = -N_r\log(\sqrt{2\pi(\sigma_r^{\text{inf}})^2})$ and $c_p = -N_r\log(\sqrt{2\pi(\sigma_r^{\text{gen}})^2} - N_s\log(\sqrt{2\pi(\sigma_s^{\text{gen}})^2}$ are constants that do not depend on network parameters. We can use these equations to evaluate our weight updates, by using the general formula in Eq. 5 and calculating derivatives. For online parameter

updates, we assume that weights change stochastically at each time step, based on samples from $\lambda_0$, $\mathbf{z}$, $\mathbf{r}$, and $\mathbf{s}$ (instead of explicitly calculating the expectation in Eq. 5):

$$\Delta \mathbf{W}^{(ij)} \propto \frac{1 - \lambda_t}{(\sigma_r^{\text{inf}})^2} (\mathbf{r}_t^{(i)} - f(\mathbf{W}\mathbf{s}_t)^{(i)}) f'(\mathbf{W}\mathbf{s}_t)^{(i)} \mathbf{s}_t^{(j)} \tag{14}$$

$$\Delta \mathbf{D}_r^{(ii)} \propto \frac{\lambda_t (1 - k_t)}{(\sigma_r^{\text{gen}})^2} (\mathbf{r}_t^{(i)} - (\mathbf{D}_r \mathbf{r}_{t-1})^{(i)}) \mathbf{r}_{t-1}^{(i)} \tag{15}$$

$$\Delta \mathbf{D}_s^{(ij)} \propto \frac{\lambda_t}{(\sigma_s^{\text{gen}})^2} (\mathbf{s}_t^{(i)} - f(\mathbf{D}_s \mathbf{r}_t)^{(i)}) f'(\mathbf{D}_s \mathbf{r}_t)^{(i)} \mathbf{r}_t^{(j)}. \tag{16}$$

Each of these updates has the form of a local synaptic plasticity rule, under the following assumptions: $\mathbf{W}^{(ij)}$ is a basal synapse from neuron $j$ to neuron $i$, $\mathbf{r}_t^{(i)}$ and $\mathbf{r}_t^{(j)}$ correspond to the pre- and post-synaptic firing rates, respectively, and $f(\mathbf{W}\mathbf{s}_t)^{(i)}$ corresponds to the local basal current injected into neuron $i$. Thus, assuming that a basal synapse has access to both the neuron's firing rate and its local basal synaptic current at a particular moment in time, $\Delta \mathbf{W}^{(ij)}$ is local; the same principle holds for the apical updates. If $\lambda_t = 0$, then network activity is driven by the generative inputs, and so the parameter updates for basal synapses depend on apically-driven activity, as has been observed experimentally [32]; similarly, apical synaptic plasticity should depend on basally-driven activity. The updates for the generative transition matrix, $\mathbf{D}_r$–determining the leakiness of the apical dendritic compartments–are gated by $1 - k_t$, indicating that parameter updates are delayed upon entering 'inference' mode: this could reasonably be implemented biologically by a slow cascade of biochemical processes that delay changes in neural parameters, as has been proposed by previous plasticity models [33, 34].

## 3   Numerical Results

**Validation on artificial stimuli**   To analyze IL performance in an environment where we have access to and control over the statistics of the latent dynamics $\mathbf{z}_t$, we constructed artificial stimuli as follows:

$$\mathbf{z}_t = \mathbf{\Lambda} \mathbf{z}_{t-1} + \sigma^{\text{true}} \boldsymbol{\eta}_t \tag{17}$$

$$\bar{\mathbf{s}}(\mathbf{z}_t) = \mathbf{A} \mathbf{z}_t, \tag{18}$$

where $\mathbf{\Lambda}$ is a $N_z \times N_z$ diagonal matrix with $\Lambda^{ii} < 1 \; \forall i$, $\mathbf{A}$ is a $N_s \times N_z$ random matrix with $A^{ij} \sim \mathcal{N}(0, \frac{1}{N_z})$, and $\boldsymbol{\eta}_t \sim \mathcal{N}(0, 1)$.[3] For simplicity, we fix the dimension of the latent space and the generative noise in the network to the ground truth values, $N_r = N_z$ neurons, and $\sigma^{\text{true}} = \sigma_r^{\text{gen}}$, so that in principle our model $\int p_m(\mathbf{r}, \mathbf{s}) d\mathbf{r}$ can match the ground truth data distribution $\int p(\mathbf{s}, \mathbf{z}) d\mathbf{z}$ exactly. This also means that we can verify that the network has learned an optimal model by comparing its second-order statistics to those of the ground truth distribution.

We trained the network using IL, verifying that the online synaptic updates minimize the loss $\mathcal{L}$ (Fig. 1c). We further validate that the network has learned to accurately perform inference, so that $q(\mathbf{r}|\mathbf{s}) \approx p_m(\mathbf{r}|\mathbf{s})$, and that the network has learned a good model of the data, so that $\int p_m(\mathbf{r}, \mathbf{s}) d\mathbf{r} \approx p(\mathbf{s})$, as per our original goals. We show that when the network is performing approximate inference, i.e. $\lambda_t = 1$, $\forall t$, stimulus reconstructions based on the network's latent state are closely matched to the actual stimuli, i.e. $\mathbf{s}_t \approx f(\mathbf{D}_s \mathbf{r}_t)$, meaning that the network is functioning as a good autoencoder across time (Fig. 1d), and across all stimulus dimensions (Fig. 1e). To verify the network's generative performance, we also show that the temporal autocorrelations for the network rates $\mathbf{r}_t$ in generative mode ($\lambda_t = 0 \; \forall t$) closely overlap with the ground truth autocorrelation structure of $\mathbf{z}$, suggesting that the learned latent features correspond (modulo a rotation) to the true latent features. Note that this latent variable match occurs because we have enforced a correspondence between the true data-generating distribution and our model, and would not necessarily happen if a different model architecture were used.

**Algorithm comparisons**   Having verified that IL is capable of training the network on simulated data, we next compared it to alternative algorithms in the literature, including neural variational

---

[3]The parameter values and initialization details for all simulations are included in the supplementary code, which was run on an internal cluster; $N_s = 100$ and $N_z = 20$.

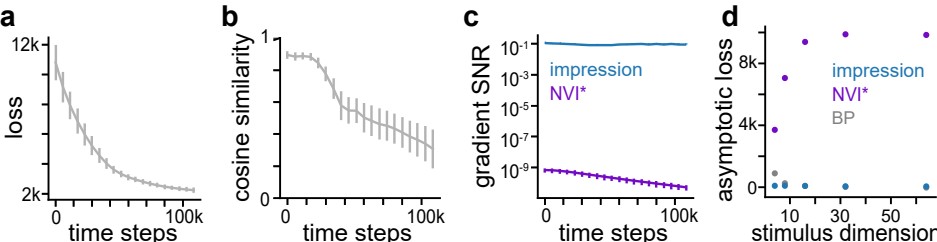

Figure 2: **Comparing learning algorithms and effects of dimensionality. a.** Loss throughout time. **b.** Cosine similarity between gradient updates given by IL and NVI*, averaged over $10^6$ samples. **c.** The signal-to-noise ratio for IL (blue), compared to NVI* (purple) across learning. **d.** Asymptotic negative ELBO loss for IL (blue), NVI* (purple), and BP (gray) as a function of the stimulus dimensionality. Error bars indicate $\pm 1$ s.e.m. averaged across 20 network realizations.

inference (NVI*), BP, and WS (see Appendix B for detailed mathematical comparisons and derivations). In particular, NVI* provides an alternative candidate model of how the brain could plausibly learn neural representations through variational inference [31]. Because NVI* performs poorly for high-dimensional stimuli and large numbers of time steps (Appendix C; [35]), we simplified the task by reducing the dimensionality of the latent space, $N_z = 2$, and stimulus space, $N_s = 4$. For twenty evenly-spaced time points over the course of the learning trajectory, we compared the inference parameter updates given by IL, $\Delta\theta_q^{\text{IL}}$, to the inference parameter updates given by NVI*, $\Delta\theta_q^{\text{NVI}}$, for a 4 time-step stimulus sequence (Fig. 2a). To get good estimates of the mean and variance of these sample parameter updates, we averaged over $10^6$ different realizations of the network noise, and compared the samples using two measures. First, we considered the cosine similarity (normalized inner product) between the two empirical mean updates, $\overline{\Delta\theta}_q^{\text{IL}} = \frac{1}{N}\sum_{k=0}^{N}\Delta\theta_q^{\text{IL}}$ and $\overline{\Delta\theta}_q^{\text{NVI}} = \frac{1}{N}\sum_{k=0}^{N}\Delta\theta_q^{\text{NVI}}$ (Fig. 2b), where $\cos(\theta) \in [-1, 1]$, and $\cos(\theta) < 0$ in this case would indicate that the parameter updates are anticorrelated. Because the NVI* update is unbiased, ie. $\mathbb{E}[\Delta\theta_q^{\text{NVI}}] = -\frac{d}{d\theta_q}\mathcal{L}$, as long as we have averaged over a sufficient number of samples $N$, a positive cosine similarity across learning between the IL update and the NVI* update (Fig. 2b) indicates that our update is aligned in expectation to the true gradient of the loss, and hence will improve performance. This is a way of empirically verifying that the bias we introduce in our derivation does not impair the learning process.

Having verified that the IL update and the true gradient are aligned on average, we next examine whether the updates given by NVI* differ in terms of their signal-to-noise ratio (SNR) from the IL updates, where we define the SNR as:

$$\text{SNR}(\Delta\theta_q) = \frac{1}{N_\theta}\sum_{i=0}^{N_\theta}\frac{\left(\overline{\Delta\theta}_q^{(i)}\right)^2}{S^2(\Delta\theta_q^{(i)})}, \qquad (19)$$

where $S^2(\cdot)$ denotes the sample variance. This measure is an average across individual parameter updates $\Delta\theta^{(i)}$, and it increases with $\left\|\overline{\Delta\theta}_q\right\|_2^2$ and decreases as the estimator variance grows. As Fig. 2c shows, the SNR is many orders of magnitude lower for NVI* than for IL over learning, likely due to the high estimator variance of the NVI*, which we demonstrate analytically for a simple example in the Appendix C. The estimator variance has direct implications for the speed of learning and asymptotic performance, so that even though NVI* and IL can have parameter updates that are aligned in expectation, due to its low variance IL will greatly outperform NVI* during training.

We verified the generality of these benefits in the same task, as we varied $N_s$, $N_z$ and $N_r$ concurrently, so that $N_s = 2N_z = 2N_r$. We optimized learning rates for NVI*, BP, and IL separately on the lowest dimensional condition by grid search across orders of magnitude ($10^{-2}$, $10^{-3}$, etc.), and found that NVI* performed worse over the entire range, while IL and BP showed similar performance (using the negative ELBO loss as a standard). Moreover, while NVI* showed worse performance as the stimulus dimension increased, this was not the case for IL or BP (Fig. 2d).

**Phase duration effects** The previous numerical results verify that IL is able to effectively learn a generative model of artificial data, and to perform inference with respect to that model. However, for

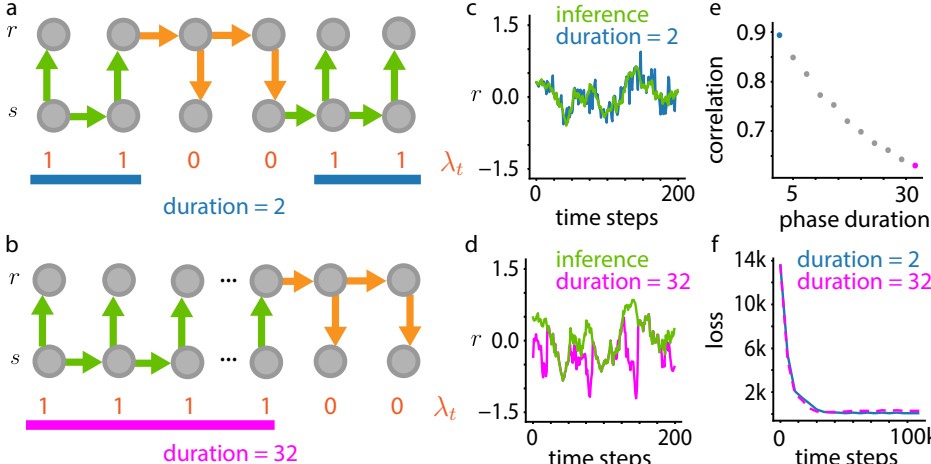

Figure 3: **The effects of phase duration on dynamics and learning. a.** Schematic of IL with a phase duration of 2 **b.** Same as a, but for a switch period of 32. **c.** Comparison of an example neuron's activity through time when the network is in inference mode (green, $\lambda_t = 1$) and when the network is alternating phase with duration 2 (blue); the random seed and stimuli are identical in both cases. **d.** Same as c, but for a phase duration of 32 (pink). **e.** The correlation across time between neurons in inference mode vs. while alternating phase, for identical random seeds. **f.** The negative ELBO loss for a network trained with a phase duration of 2 (blue, solid line) or 32 (pink, dashed line).

IL to be a valid candidate for online learning in the brain, the learning process should not significantly interfere with perception. To test this, we explored how the 'phase duration' $K$ affects the correlation between network activity in a simulation where $\lambda_t = 1$, $\forall t$, and a simulation where $\lambda_t$ alternates phases every $K$ time steps (for a fixed random seed and stimulus sequence). If the learning process did not interfere with perception at all, this correlation would be 1, and if it completely disrupted perception it would be 0, or even negative. In Fig. 3c and d, we show two example traces with $K = 2$ and $K = 32$, respectively, comparing the network in inference mode to the network during learning. While neural trajectories for the shorter phase durations are closely correlated, they deviate considerably for longer phase durations (Fig. 3c-e). Despite this, the loss profile (negative ELBO) is identical. Since WS can be viewed as a special case of IL for very long phase durations (Appendix B.3; see Fig. S1a for an even longer phase duration), this implies that the two methods have similar performance. However, IL operating in a mode of fast fluctuations between inference and generation may be more biologically relevant, as this reduces the interference with perception without impairing learning. Moreover, we found that lengthening the duration of the inference phase alone while keeping very short bursts of generative activity further reduced perceptual disturbance, while only slightly increasing the time required to learn (Fig. S1b-d).

**Spoken digits task** Having verified the performance of IL on artificial stimuli, we next tested its performance on higher-dimensional and more complex naturalistic stimuli. We used the training and test sets of the Free Spoken Digits Dataset [36], which provides audio time series of humans speaking digits 0-9.[4] We transformed these time series into log-mel spectrograms as a coarse approximation of the initial stages of the human auditory system, shifted the inputs by a constant so as to make them all positive, and divided the result by the across-channel standard deviation. The results of Fig. 4 are shown in the original log-mel spectrogram input space.

To assess the hierarchical processing capabilities of IL, we added an additional feedforward layer to the network architecture (Fig. 4a); we provide the details of how this modification affects simulation and parameter updates in Appendix D. To compare IL to NVI*, we again optimized learning rates via grid search across orders of magnitude, and found that IL greatly outperformed NVI* when each was evaluated at its respective optimal learning rate (Fig. 4b). Furthermore, we observed that our

---

[4]The FSDD is available at `https://github.com/Jakobovski/free-spoken-digit-dataset`.

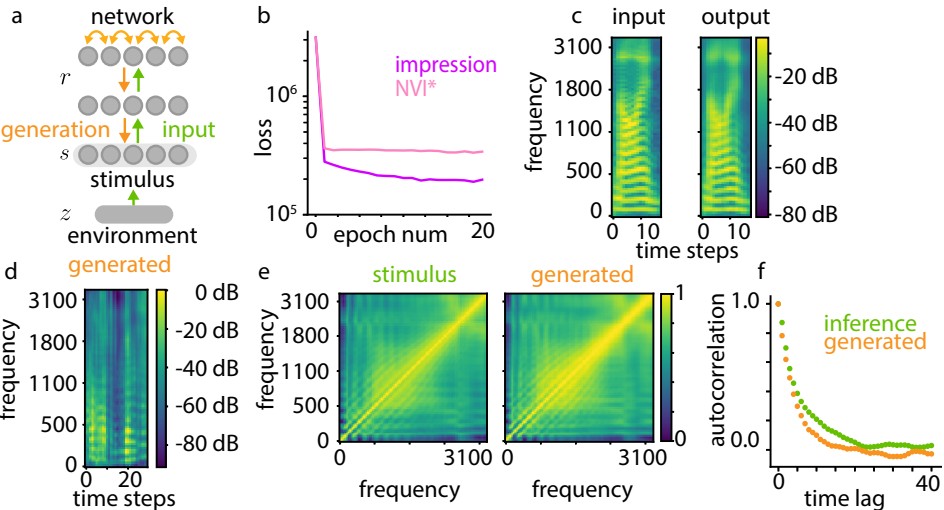

Figure 4: **Learning auditory sequences in a multilayer network. a.** Hierarchical network architecture. **b.** Test loss across epochs for IL (blue) and NVI* (purple). **c.** Comparison between an example data input and the corresponding network output in inference mode ($\lambda_t = 1$). **d.** Sample network output in generative mode ($\lambda_t = 0$). **e.** Across-frequency amplitude correlations for the data (left) and for network-generated samples (right). **f.** Auto-correlation function of a neuron in inference and generative modes.

trained network meets the same criteria for success as for our artificial stimuli, namely its stimulus reconstructions closely match the true stimulus while in inference mode ($\lambda_t = 1 \,\forall t$; Fig. 4c), and sample stimuli produced while the network is in generative mode ($\lambda_t = 0 \,\forall t$) qualitatively correspond to ground-truth stimuli (Fig. 4d), and quantitatively match the structure of both spatial (Fig. 4e) and temporal (Fig. 4f) autocorrelation of the input. These results collectively demonstrate that IL is capable of training neural representations of complex real-world stimuli. They also show that IL can function when there is a mismatch between its architecture and the structure of environmental latent variables, which are in this case unknown. In general, learning may fail if the chosen network architecture is too restrictive.

## 4  Discussion

Impression learning (IL) provides a potential mechanism for the brain to learn generative models of its sensory inputs through local synaptic plasticity, while concurrently performing approximate inference with respect to these models. IL is a direct generalization of the Wake-Sleep algorithm [29], which replaces lengthy offline 'Sleep' phases with brief substitutions of network-generated samples in place of incoming data, in a way that minimally perturbs natural neural trajectories. Transitions between 'inference mode' and 'generative mode' are controlled by a global signal $\lambda_t$, which decides whether generative signals to the apical synapses or inference signals to the basal synapses dominate network activity.

Computationally, IL outperforms NVI* [30, 31], a particular instance of three-factor plasticity [37], because its internal model provides explicit 'credit assignment' for each individual neuron, rather than implicitly calculating it via correlations between neural activity and a global reward signal. This leads to lower-variance gradient estimates and faster learning. Alternative learning algorithms such as backpropagation (through time) [38] are not intrinsically probabilistic, but can be used for optimizing probabilistic objectives. Like IL, BP provides explicit credit assignment, but the parameter updates it provides are nonlocal across both network layers and time. It is worth noting that IL was developed in a purely unsupervised learning setting, whereas both BP and NVI* extend to supervised and reinforcement learning [39, 40]. In the context of supervised learning, several biologically-plausible approximations to BP leverage the apical-basal dendritic structure of pyramidal neurons to learn

[25, 21], based primarily on target-propagation [41] or its variants [42]. It would be valuable to explore the combination of such extensions with the continuous online learning capabilities of IL.

Local computations are considered a necessary condition for learning algorithms to be biologically-plausible. In our framework, locality is enforced through the structure of the internal graphical model ($p_m$) and the approximate inference distribution ($q$): any choice of neural network architecture with independent noise will guarantee local plasticity. Our framework is relatively agnostic to the details: neurons could be either rate-based with Gaussian intrinsic noise (as in the examples presented here), or generate spikes with Poisson variability, which would result in synaptic updates analogous to empirically observed spike-timing-dependent plasticity, as found in generalizations of WS [14]. It would also be possible to make distinctions between excitatory and inhibitory neurons, by requiring all outgoing synapses from individual neurons to be either positive or negative, or to include more complex dendritic arborizations, as have been explored in recent experimental [43] and modeling [44] efforts. Our current model enforces hard, global phase distinctions ($\lambda_t \in \{0, 1\}$ for all neurons), which could potentially correspond to alternations between activity driven by apical dendritic calcium events and basal spiking tied to theta oscillations in the hippocampus [32]. However, cortical data indicate that input to apical and basal dendrites contribute concurrently and constructively to spiking activity [45]. We are currently working to extend our derivation to these circumstances, by allowing $\lambda_t$ to be non-binary and heterogenous across neurons.

Traditional predictive coding [7] requires steady-state assumptions for learning, meaning that neural dynamics must occur on a timescale much faster than that of stimuli. In contrast, IL requires a mechanism by which the relative influence of the apical and basal dendrites of pyramidal neurons can be rapidly switched, along with learning mechanisms that operate at that timescale. If such a mechanism could be experimentally identified and controlled, our model makes the specific prediction that increasing the dominance of apical dendritic input on neural activity ($\lambda_t \approx 1$) would cause the network to sample from its generative model, i.e. the manipulation will induce structured hallucinations that mimic realistic stimuli (and associated neural activity), without being tied to the sensory world. One candidate gating mechanism is rapid inhibition targeting apical dendrites specifically [46–49]; but much work remains to explicitly relate this mechanism to learning and plasticity.

IL predicts that synapses will use an error signal based on the difference between local dendritic compartmental currents (either apical or basal) and the neuron's total firing rate to perform learning. There is some evidence that spiking activity driven by apical inputs to pyramidal neurons can induce plasticity at basal synapses [32, 50], and several studies have found systematic changes in synaptic plasticity between apical and basal synapses, in particular the sign changes induced by local dendritic inputs that IL predicts [51–54]. Hence, IL has the potential to explain the diversity of plasticity phenomena observed experimentally and inform future experiments.

## Acknowledgments and Disclosure of Funding

We thank Camille Rullán Buxó, Caroline Haimerl, Owen Marschall, Pedro Herrero-Vidal, Siavash Golkar, David Lipshutz, Yanis Bahroun, Tiberiu Tesileanu, Eilif Muller, Blake Richards, Guillaume Lajoie, Maximilian Puelma Touzel, and Alexandre Payeur for helpful discussions and feedback on earlier versions of this manuscript. We gratefully acknowledge the Howard Hughes Medical Institute and the Simons Foundation for their support of this work. CS is supported by National Institute of Mental Health Award 1R01MH125571-01, by the National Science Foundation under NSF Award No.1922658 and a Google faculty award.

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
