# Impression learning: Online representation learning with synaptic plasticity
## –Appendices–

**Colin Bredenberg**
Center for Neural Science
New York University
cjb617@nyu.edu

**Benjamin S. H. Lyo**
Center for Neural Science
New York University
blyo@nyu.edu

**Eero P. Simoncelli**
Center for Neural Science,
New York University
Flatiron Institute, Simons Foundation
eero.simoncelli@nyu.edu

**Cristina Savin**
Center for Neural Science,
Center for Data Science
New York University
csavin@nyu.edu

## A   Bias calculation

Our derivation of the update for IL (Eq. 3) is based on an expansion of $\log \frac{\tilde{p}_\theta}{\tilde{q}_\theta}$ about $\frac{\tilde{p}_\theta}{\tilde{q}_\theta} = 1$:

$$\int \left[ \log \frac{\tilde{p}_\theta}{\tilde{q}_\theta} \right] (\nabla_\theta \log \tilde{q}_\theta) \tilde{q}_\theta \; d\mathbf{r}d\mathbf{s} = \int \left[ \frac{\tilde{p}_\theta}{\tilde{q}_\theta} - 1 \right] (\nabla_\theta \log \tilde{q}_\theta) \tilde{q}_\theta \; d\mathbf{r}d\mathbf{s} \tag{S1}$$

$$- \frac{1}{2} \int \left[ \frac{(\frac{\tilde{p}_\theta}{\tilde{q}_\theta} - 1)}{1 + \epsilon(\mathbf{r}, \mathbf{s})} \right]^2 (\nabla_\theta \log \tilde{q}_\theta) \tilde{q}_\theta \; d\mathbf{r}d\mathbf{s},$$

for some $\epsilon(\mathbf{r}, \mathbf{s})$ st. $|\epsilon(\mathbf{r}, \mathbf{s})| < |\frac{\tilde{p}}{\tilde{q}} - 1|$. Note that this is not a truncated Taylor series approximation – we are instead using Taylor's theorem, and the second term provides an exact expression for the bias. We can use the Caucy-Schwartz inequality for expectations to bound this as follows:

$$|\text{bias}| = \frac{1}{2} \left| \int \left[ \frac{(\frac{\tilde{p}_\theta}{\tilde{q}_\theta} - 1)}{1 + \epsilon(\mathbf{r}, \mathbf{s})} \right]^2 (\nabla_\theta \log \tilde{q}_\theta) \tilde{q}_\theta \; d\mathbf{r}d\mathbf{s} \right|$$

$$\leq \frac{1}{2} \sqrt{\int \left[ \frac{(\frac{\tilde{p}_\theta}{\tilde{q}_\theta} - 1)}{1 + \epsilon(\mathbf{r}, \mathbf{s})} \right]^4 \tilde{q}_\theta \; d\mathbf{r}d\mathbf{s}} \sqrt{\int (\nabla_\theta \log \tilde{q}_\theta)^2 \tilde{q}_\theta \; d\mathbf{r}d\mathbf{s}}, \tag{S2}$$

We examine the consequences of this bias formula for our specific model. Consider the component of the gradient with respect to the feedforward weight $\mathbf{W}^{(ij)}$:

$$\frac{d}{dW^{(ij)}} \log \tilde{q}_\theta = \sum_t \frac{\lambda_t}{(\sigma_r^{\text{inf}})^2} (\mathbf{r}_t^{(i)} - f(\mathbf{W}\mathbf{s}_t)^{(i)}) f'(\mathbf{W}\mathbf{s}_t)^{(i)} \mathbf{s}_t^{(j)}.$$

Note that $f(\cdot) < 1$ and $f'(\cdot) < 1$ for the $\tanh$ function, and assume that $(s_t^{(j)})^2 < S \;\; \forall t$ for some constant $S$. Defining $B = \sqrt{\int \left[ \frac{(\frac{\tilde{p}_\theta}{\tilde{q}_\theta} - 1)}{1 + \epsilon(\mathbf{r}, \mathbf{s})} \right]^4 \tilde{q}_\theta \; d\mathbf{r}d\mathbf{s}}$, and substituting the gradient component gives:

35th Conference on Neural Information Processing Systems (NeurIPS 2021).

$$|\text{bias}| \le \frac{B}{2}\sqrt{\int\left(\sum_t \frac{\lambda_t}{(\sigma_r^{\text{inf}})^2}(\mathbf{r}_t^{(i)} - f(\mathbf{W}\mathbf{s}_t)^{(i)})f'(\mathbf{W}\mathbf{s}_t)^{(i)}\mathbf{s}_t^{(j)}\right)^2 \tilde{q}_\theta\, d\mathbf{r}d\mathbf{s}}$$

$$= \frac{B}{2}\sqrt{\int \sum_t \sum_{t'} \frac{\lambda_t \lambda_{t'}}{(\sigma_r^{\text{inf}})^4}(\mathbf{r}_t^{(i)} - f(\mathbf{W}\mathbf{s}_t)^{(i)})(\mathbf{r}_{t'}^{(i)} - f(\mathbf{W}\mathbf{s}_{t'})^{(i)})f'(\mathbf{W}\mathbf{s}_t)^{(i)}f'(\mathbf{W}\mathbf{s}_{t'})^{(i)}\mathbf{s}_t^{(j)}\mathbf{s}_{t'}^{(j)}\tilde{q}_\theta\, d\mathbf{r}d\mathbf{s}}$$

$$= \frac{B}{2}\sqrt{\int \sum_t \frac{\lambda_t^2}{(\sigma_r^{\text{inf}})^4}(\mathbf{r}_t^{(i)} - f(\mathbf{W}\mathbf{s}_t)^{(i)})^2(f'(\mathbf{W}\mathbf{s}_t)^{(i)}\mathbf{s}_t^{(j)})^2\tilde{q}_\theta\, d\mathbf{r}d\mathbf{s}},$$

where this second equality follows from the fact that $\mathbf{r}_t^{(i)} - f(\mathbf{W}\mathbf{s}_t)^{(i)} \sim \mathcal{N}(0, \sigma_r^{\text{inf}})$ without any temporal correlation, so that $\mathbb{E}\left[(\mathbf{r}_t^{(i)} - f(\mathbf{W}\mathbf{s}_t)^{(i)})(\mathbf{r}_{t'}^{(i)} - f(\mathbf{W}\mathbf{s}_{t'})^{(i)})\right]_{\mathbf{r}|\mathbf{s}} = 0$ for $t \ne t'$. Continuing our derivation, we have:

$$|\text{bias}| \le \frac{B}{2}\sqrt{\sum_t \frac{\lambda_t^2}{(\sigma_r^{\text{inf}})^4}\int(\mathbf{r}_t^{(i)} - f(\mathbf{W}\mathbf{s}_t)^{(i)})^2(f'(\mathbf{W}\mathbf{s}_t)^{(i)}\mathbf{s}_t^{(j)})^2\tilde{q}_\theta(\mathbf{r},\mathbf{s})\, d\mathbf{r}d\mathbf{s}}$$

$$= \frac{B}{2}\sqrt{\sum_t \frac{\lambda_t^2}{(\sigma_r^{\text{inf}})^2}\int(f'(\mathbf{W}\mathbf{s}_t)^{(i)}\mathbf{s}_t^{(j)})^2\tilde{q}_\theta(\mathbf{s})\, d\mathbf{s}}$$

$$\le \frac{B}{2}\sqrt{\frac{S}{(\sigma_r^{\text{inf}})^2}\sum_t \lambda_t^2}$$

$$= \frac{B}{2}\sqrt{\frac{ST}{2(\sigma_r^{\text{inf}})^2}}, \tag{S3}$$

where $T$ is the total time. Thus, for our particular choice of neural model, the bias is proportional to $B$, which vanishes as performance improves. Note that the update term in Eq. (S1) is $\mathcal{O}(|\frac{\tilde{p}}{\tilde{q}} - 1|)$, so its magnitude is expected to be much larger than the bias in the vicinity of a global optimum. The $\sqrt{T/(\sigma_r^{\text{inf}})^2}$ proportionality constant also should not be a cause for concern: the gradient itself scales with $T/(\sigma_r^{\text{inf}})^2$, and thus small values of $(\sigma_r^{\text{inf}})^2$ will not make the relative error explode.

## B  Comparison to other algorithms

In this section, we explore the relationships between impression learning (IL) and other stochastic learning algorithms. Specifically, we consider a variant of neural variational inference (NVI*), backpropagation (BP), and Wake-Sleep (WS).

### B.1  Neural Variational Inference

Neural variational inference is a learning algorithm for neural networks that optimizes the evidence lower bound (ELBO) objective function. Here, we modify the algorithm by incorporating our novel loss (Eq. 2), producing a variant that we call NVI*. We first take the derivative of our loss, without approximations. These steps are identical to the initial steps in our derivation of IL, up to the Taylor expansion:

$$-\nabla_\theta \mathcal{L} = -\nabla_\theta \mathbb{E}_{\lambda,\mathbf{z}}\left[\int [\log \tilde{q}_\theta - \log \tilde{p}_\theta]\, \tilde{q}_\theta\, d\mathbf{r}d\mathbf{s}\right]$$

$$= -\mathbb{E}_{\lambda,\mathbf{z}}\left[\int [\nabla_\theta(\log \tilde{q}_\theta - \log \tilde{p}_\theta)]\, \tilde{q}_\theta\, d\mathbf{r}d\mathbf{s} + \int [\log \tilde{q}_\theta - \log \tilde{p}_\theta]\, \nabla_\theta \tilde{q}_\theta\, d\mathbf{r}d\mathbf{s}\right]$$

$$= -\mathbb{E}_{\lambda,\mathbf{z}}\left[\int [\nabla_\theta \log \tilde{q}_\theta - \nabla_\theta \log \tilde{p}_\theta]\, \tilde{q}_\theta\, d\mathbf{r}d\mathbf{s} + \int [\log \tilde{q}_\theta - \log \tilde{p}_\theta]\, (\nabla_\theta \log \tilde{q}_\theta)\tilde{q}_\theta\, d\mathbf{r}d\mathbf{s}\right]$$

$$= \mathbb{E}_{\lambda,\mathbf{z}}\left[\int [\nabla_\theta \log \tilde{p}_\theta]\, \tilde{q}_\theta\, d\mathbf{r}d\mathbf{s} + \int \left[\log \frac{\tilde{p}_\theta}{\tilde{q}_\theta}\right](\nabla_\theta \log \tilde{q}_\theta)\tilde{q}_\theta\, d\mathbf{r}d\mathbf{s}\right] \tag{S4}$$

Updates calculated by these samples will be unbiased in expectation, because there are no approximations. However, we will show in Appendix C that these updates may have high variance.

To provide a fair comparison to IL, we have added two additional features that have been shown to reduce the variance of sample estimates [1, 2]. The first involves subtracting a control variate from our second term:

$$-\nabla_\theta \mathcal{L} = \mathbb{E}_{\lambda, \mathbf{z}} \left[ \int \left[ \nabla_\theta \log \tilde{p}_\theta \right] \tilde{q}_\theta \, d\mathbf{r} d\mathbf{s} + \int \left( \log \frac{\tilde{p}_\theta}{\tilde{q}_\theta} - \mathbb{E}\left[ \log \frac{\tilde{p}_\theta}{\tilde{q}_\theta} \right] \right) (\nabla_\theta \log \tilde{q}_\theta) \tilde{q}_\theta \, d\mathbf{r} d\mathbf{s} \right]. \quad \text{(S5)}$$

The subtracted term, $\mathbb{E}\left[ \log \frac{\tilde{p}_\theta}{\tilde{q}_\theta} \right] \int (\nabla_\theta \log \tilde{q}_\theta) \tilde{q}_\theta \, d\mathbf{r} d\mathbf{s}$, is zero because it is a constant times the expectation of the score function. As such, it keeps the weight updates unbiased, but can still significantly reduce the variance.

The original NVI method employs a dynamic baseline estimated with a neural network as a function of inputs $\mathbf{s}$. It is likely that this more flexible control variate can further reduce the variance of parameter estimates beyond the baseline that we explore here. However, this baseline was trained with backpropagation, and as such, would not provide a biologically-plausible comparison. We can approximate Eq. S5 by summing over samples from $\tilde{q}_\theta$, and updating our weights at every time point:

$$\Delta\theta \propto \left[ \nabla_\theta \log \tilde{p}_t(\mathbf{r}_t, \mathbf{s}_t; \theta) \right] + \left[ \log \frac{\tilde{p}_t}{\tilde{q}_t} - \bar{\mathcal{L}} \right] \sum_{s=0}^{t} (\nabla_\theta \log \tilde{q}_t(\mathbf{r}_t, \mathbf{s}_t; \theta))$$

$$\propto \left[ \nabla_\theta \log \tilde{p}_t(\mathbf{r}_t, \mathbf{s}_t; \theta) \right] + \left[ \log \frac{\tilde{p}_t}{\tilde{q}_t} - \bar{\mathcal{L}} \right] g^\theta, \quad \text{(S6)}$$

where $\bar{\mathcal{L}}$ is approximated online according to a running average of the loss at each time step, and $g^\theta$, called an 'eligibility trace' [3], is computed by a running integral. These quantities are both computed online as follows:

$$\bar{\mathcal{L}}_t = \gamma_\mathcal{L} \log \frac{\tilde{p}_t}{\tilde{q}_t} + (1 - \gamma_\mathcal{L})\bar{\mathcal{L}}_{t-1} \quad \text{(S7)}$$

$$g_t^\theta = \nabla_\theta \log \tilde{q}_t(\mathbf{r}_t, \mathbf{s}_t; \theta) + \gamma_g g_{t-1}^\theta, \quad \text{(S8)}$$

where $\gamma_\mathcal{L} \ll 1$, so that $\bar{\mathcal{L}}_t$ is a weighted average of past losses. If we want an unbiased estimate of the gradient, then we would take $\gamma_g = 1$, so that $g_t^\theta = \sum_{s=0}^{t}(\nabla_\theta \log \tilde{q}_t(\mathbf{r}_t, \mathbf{s}_t; \theta))$. However, the variance of this eligibility trace grows without bound as $T \to \infty$, which makes online learning using this algorithm nearly impossible without approximation. For this reason, we take $\gamma_e$ as a constant less than, but close to 1 when we compare NVI* to IL performance, which introduces a small bias, with the benefit of allowing for online learning. This is a technique commonly employed in the three-factor plasticity literature [4, 5], and can be thought of as an analog to temporal windowing in backpropagation through time [6]. For our numerical gradient comparisons (Fig. 2), however, we used a short number of time steps, but took $\gamma_g = 1$ to remove all bias.

This method of differentiation is particularly important to compare to IL, because it can be thought of as a three-factor synaptic plasticity rule, where for a neural network, the parameter update becomes a global 'loss' signal $\log \frac{\tilde{p}_t}{\tilde{q}_t} - \bar{\mathcal{L}}$ combined with synaptically local terms $g^\theta$ and $\nabla_\theta \log \tilde{p}_t(\mathbf{r}_t, \mathbf{s}_t; \theta)$, the second of which comprises the entirety of the IL update. Typically for reinforcement learning, the global 'reward' signal is justified by referencing neuromodulatory signals that project broadly to synapses throughout the cortex and carry information about reward [7, 4, 8, 9]. However, the origins of the global 'loss' in our *unsupervised* case are unclear. Furthermore, as we show in Appendix C, the term $\left[ \log \frac{\tilde{p}_t}{\tilde{q}_t} - \bar{\mathcal{L}} \right] g^\theta$ is high variance, and requires orders of magnitude more samples (or lower learning rates) in order to get a useful gradient estimate. A technical way of viewing our contribution in this paper is that we have shown that the $\left[ \log \frac{\tilde{p}_t}{\tilde{q}_t} - \bar{\mathcal{L}} \right] g^\theta$ term is largely redundant and unnecessary for effective learning on our unsupervised objective, and that discarding it produces substantial performance increases while allowing the parameter update to remain a completely local synaptic plasticity rule for neural networks.

## B.2 Backpropagation

Backpropagation (BP) cannot be performed for stochastic variables $\mathbf{r}_t$, because under an expectation, these are integration variables with no explicit dependency on any parameters. For this reason, when

computing a derivative of our loss using NVI*, we differentiate the *probability distribution*, which depends on network parameters. However, as we will show below, this straightforward method can result in high variance parameter estimates. The classical alternative to NVI* is to perform the 'reparameterization trick,' in which a change of variables allows the use of stochastic gradient descent with BP. This trick is largely responsible for the success of the variational autoencoder [10, 11], though it is well known that BP does not produce synaptically local parameter updates. Here, we use BP as an upper bound for comparison, with the understanding that local learning algorithms are unlikely to be able to completely match its performance. Below, we review its calculation, starting with changing our variable of integration.

It is worth noting that this 'reparameterization' will work only for additive Gaussian noise. As such, applying BP to our network will only be possible for a restricted set of noise models, and can fail in particular for Poisson-spiking network models, where IL, NVI*, and WS will not. For each time point, we define $\boldsymbol{\eta}_t = \mathbf{r}_t - \bar{\mathbf{r}}_t^q(\theta, \lambda, \boldsymbol{\eta}_{0:t-1}, \boldsymbol{\xi}_{0:t-1})$, where $\bar{\mathbf{r}}_t^q(\theta, \lambda, \boldsymbol{\eta}_{0:t-1}, \boldsymbol{\xi}_{0:t-1})$ is the mean firing rate conditioned on noise, stimulus, and $\lambda$ values from previous time steps (given by $\tilde{q}$). Similarly, we define $\boldsymbol{\xi}_t = \mathbf{s}_t - \bar{\mathbf{s}}_t^q(\theta, \lambda, \boldsymbol{\eta}_{0:t-1}, \boldsymbol{\xi}_{0:t-1})$. This defines $\boldsymbol{\eta}_t$ and $\boldsymbol{\xi}_t$ as the noise added on top of every firing rate and stimulus at time $t$. Instead of integrating over the rates and stimuli, we integrate over these fluctuations, replacing each instance of $\mathbf{r}_t$ with $\bar{\mathbf{r}}_t^q(\theta, \lambda, \boldsymbol{\eta}_{0:t-1}, \boldsymbol{\xi}_{0:t-1}) + \boldsymbol{\eta}_t$ and $\mathbf{s}_t$ with $\bar{\mathbf{s}}_t^q(\theta, \lambda, \boldsymbol{\eta}_{0:t-1}, \boldsymbol{\xi}_{0:t-1}) + \boldsymbol{\xi}_t$. We will refer to the mean parameters of $\tilde{p}_\theta$ where these substitutions have been made as $\bar{\mathbf{r}}_t^p(\theta, \lambda, \boldsymbol{\eta}_{0:t-1}, \boldsymbol{\xi}_{0:t-1})$ and $\bar{\mathbf{s}}_t^q(\theta, \lambda, \boldsymbol{\eta}_{0:t-1}, \boldsymbol{\xi}_{0:t-1})$. Our new random variables have the probability distributions: $p(\boldsymbol{\eta}_t) = \mathcal{N}(0, \lambda_t \sigma_r^{inf} + (1 - \lambda_t) \sigma_r^{\mathrm{gen}})$ and $p(\boldsymbol{\xi}_t) = \mathcal{N}(0, \lambda_t \sigma_s^{\mathrm{inf}} + (1 - \lambda_t) \sigma_s^{\mathrm{gen}})$. Performing our change of variables gives:

$$-\nabla_\theta \mathcal{L} = -\nabla_\theta \int \left[ \log \tilde{q}_\theta - \log \tilde{p}_\theta \right] \tilde{q}_\theta \, d\mathbf{r} d\mathbf{s}$$

$$= -\nabla_\theta \int \left[ \log \prod_t \frac{1}{Z} \exp\left( \frac{-\boldsymbol{\eta}_t^2}{2(\lambda_t \sigma_s^{\mathrm{inf}} + (1 - \lambda_t) \sigma_s^{\mathrm{gen}})^2} \right) \right] p(\boldsymbol{\eta}, \boldsymbol{\xi}) \, d\boldsymbol{\eta} d\boldsymbol{\xi}$$

$$- \nabla_\theta \int \left[ \log \prod_t \frac{1}{Z} \exp\left( \frac{-\boldsymbol{\xi}_t^2}{2(\lambda_t \sigma_s^{\mathrm{inf}} + (1 - \lambda_t) \sigma_s^{\mathrm{gen}})^2} \right) \right] p(\eta, \xi) \, d\boldsymbol{\eta} d\boldsymbol{\xi}$$

$$+ \nabla_\theta \int \left[ \log \prod_t \frac{1}{Z} \exp\left( \frac{-(\bar{\mathbf{r}}_t^q + \boldsymbol{\eta}_t - \bar{\mathbf{r}}_t^p)^2}{2((1 - \lambda_t) \sigma_r^{\mathrm{inf}} + \lambda_t \sigma_r^{\mathrm{gen}})^2} \right) \right] p(\boldsymbol{\eta}, \boldsymbol{\xi}) \, d\boldsymbol{\eta} d\boldsymbol{\xi}$$

$$+ \nabla_\theta \int \left[ \log \prod_t \frac{1}{Z} \exp\left( \frac{-(\bar{\mathbf{s}}_t^q + \boldsymbol{\xi}_t - \bar{\mathbf{s}}_t^p)^2}{2((1 - \lambda_t) \sigma_s^{\mathrm{inf}} + \lambda_t \sigma_s^{\mathrm{gen}})^2} \right) \right] p(\boldsymbol{\eta}, \boldsymbol{\xi}) \, d\boldsymbol{\eta} d\boldsymbol{\xi}$$

$$= \mathbb{E}_{\boldsymbol{\eta}, \boldsymbol{\xi}} \left[ \nabla_\theta \sum_t -\frac{(\bar{\mathbf{r}}_t^q(\theta, \boldsymbol{\eta}, \boldsymbol{\xi}) + \boldsymbol{\eta}_t - \bar{\mathbf{r}}_t^p(\theta, \boldsymbol{\eta}, \boldsymbol{\xi}))^2}{2((1 - \lambda_t) \sigma_r^{\mathrm{inf}} + \lambda_t \sigma_r^{\mathrm{gen}})^2} - \frac{(\bar{\mathbf{s}}_t^q(\theta, \boldsymbol{\eta}, \boldsymbol{\xi}) + \boldsymbol{\xi}_t - \bar{\mathbf{s}}_t^p(\theta, \boldsymbol{\eta}, \boldsymbol{\xi}))^2}{2((1 - \lambda_t) \sigma_s^{\mathrm{inf}} + \lambda_t \sigma_s^{\mathrm{gen}})^2} \right],$$

$$\text{(S9)}$$

where the last equality follows from the fact that $\eta_t$ and $\xi_t$ have no dependence on the network parameters. Now, the parameter dependence is contained in $\bar{\mathbf{r}}_t^q$, $\bar{\mathbf{r}}_t^p$, $\bar{\mathbf{s}}_t^q$, and $\bar{\mathbf{s}}_t^p$, all of which depend on the mean firing rates at *each previous time step*: using BP to compute the gradients of these mean parameters leads to nonlocal updates, which is the key reason BP is a biologically-implausible algorithm [12]. For our simulations, we set $\lambda_t = 1 \, \forall t$, so that our parameter updates were equivalent to minimizing the negative ELBO, and gradients were computed using Pytorch [13]. In subsequent sections, we will show that weight updates computed using samples from this expectation will generally have much lower variance than NVI*.

## B.3 Wake-Sleep

As already mentioned, WS can be viewed as a special case of IL. To show this, we can take $\lambda_t = \lambda_0 \, \forall t$, with $p(\lambda_0 = 0) = p(\lambda_0 = 1) = 0.5$ (for IL, $\lambda_t$ alternates with phase duration $K = 2$). For this

choice of $\lambda$, we follow our IL derivation (Eq. 5), and get:

$$-\nabla_\theta \mathcal{L} \approx 2\mathbb{E}_{\lambda_0,\mathbf{z}} \left[ \int \left[ \sum_t (1-\lambda_t)\nabla_\theta \log q_t + (\lambda_t)\nabla_\theta \log p_{mt} \right] \tilde{q}_\theta \, d\mathbf{r}d\mathbf{s} \right]$$

$$= \mathbb{E}_{\mathbf{z}} \left[ \int \left[ \sum_t \nabla_\theta \log q_t \right] p_m(\mathbf{r},\mathbf{s}) \, d\mathbf{r}d\mathbf{s} + \int \left[ \sum_t \nabla_\theta \log p_{mt} \right] q(\mathbf{r}|\mathbf{s})p(\mathbf{s}|\mathbf{z}) \, d\mathbf{r}d\mathbf{s} \right].$$

$$\text{(S10)}$$

Since WS is a special case of IL, the bias properties of its individual samples are identical. However, typically WS weight updates are computed coordinate-wise, updating parameters for $p_m$ and $q$ separately, whose updates are computed after averaging over many samples. This can lead to behavior that approximates the EM algorithm under restrictive conditions, a fact that is used in the proofs of convergence of the WS algorithm for simple models [14]. Because our algorithm does not perform coordinate descent, it is best viewed as an approximation to gradient descent with a well-behaved bias, rather than an approximation of the EM algorithm.

The WS parameter updates can also be interpreted as synaptic plasticity at apical and basal dendrites of pyramidal neurons, as with IL. The key difference is that WS requires lengthy phases where $\lambda_t = 1 \; \forall t$ (Wake) and where $\lambda_t = 0 \; \forall t$ (Sleep). The requirement that the network remain in a generative state while training the inference parameters $\theta_q$ would require a biological organism to explicitly hallucinate while training its parameters. Though such generative states may be possible in some restricted form, and WS could perfectly coexist with IL in a biological organism, we believe the more general perspective afforded by IL is much more likely to correspond to biology than the phase distinctions required by WS. The benefits to perceptual continuity given by IL over WS come from its ability to leverage temporal predictability in both network states and stimuli by only staying in a generative state for a brief period of time. However, for static images and neural architectures, IL and WS are much more similar, effectively amounting to different schedules for updating generative and inference parameters in alternating sequence.

## C   Estimator variance

In Appendix A, we explored the bias introduced by the approximations used in the derivation of IL. Here, we consider the variance of sample weight updates, and compare to the variability of samples obtained from more standard methods, in particular BP and NVI*, whose sampling-based estimates have can have very different variances [11].

To keep the analysis tractable, we will study a simple example: maximizing our modified KL divergence between two time series composed of temporally-uncorrelated univariate normal distributions with identical variance and different means: $p(r_t) \sim \mathcal{N}(\mu_p, \sigma^2)$, $q(r_t) \sim \mathcal{N}(\mu_q, \sigma^2)$. We define $\lambda_t$ such that $p(\lambda_t = 0) = p(\lambda_t = 1) = 0.5 \; \forall t$. This produces the two hybrid distributions:

$$\tilde{p}(r|\lambda_t) = \prod_{t=0}^{T} p(r_t)^{\lambda_t} q(r_t)^{(1-\lambda_t)} \tag{S11}$$

$$\tilde{q}(r|\lambda_t) = \prod_{t=0}^{T} p(r_t)^{(1-\lambda_t)} q(r_t)^{\lambda_t}. \tag{S12}$$

Using these hybrid distributions, we can write our objective function as:

$$\mathcal{L} = \mathbb{E}_{\lambda_t} \left[ KL(\tilde{q}||\tilde{p}) \right] = \int \left[ \int (\log \tilde{q}(r|\lambda_t) - \log \tilde{p}(r|\lambda_t))\tilde{q}(r|\lambda_t)dr \right] p(\lambda_t)d\lambda_t. \tag{S13}$$

We will show that our three methods: NVI*, BP, and IL (which here will coincide exactly with WS), all produce unbiased stochastic gradient estimates, with very different variance properties.

It is worth explicitly outlining why variance is such an important quantity for stochastic gradient estimates. Suppose we obtain $N$ independent samples of a weight update $\Delta\mu_q$, and want to compute

the MSE of our estimated weight update to the *true* gradient, in expectation over samples:

$$MSE(\Delta\mu_q) = \mathbb{E}_{\Delta\mu_q^{(n)}}\left[\left(-\frac{d}{d\mu_q}\mathcal{L} - \frac{1}{N}\sum_{n=0}^{N}\Delta\mu_q^{(n)}\right)^2\right]$$

$$= \left(-\frac{d}{d\mu_q}\mathcal{L} - \mathbb{E}_{\Delta\mu_q^{(n)}}\left[\frac{1}{N}\sum_{n=0}^{N}\Delta\mu_q^{(n)}\right]\right)^2 + Var\left[\frac{1}{N}\sum_{n=0}^{N}\Delta\mu_q^{(n)}\right]. \quad \text{(S14)}$$

Here, the equality follows from bias-variance decomposition of the mean-squared error. In our toy example (but not in general) the biases for IL, BP, and NVI* will all be 0. This gives:

$$MSE(\Delta\mu_q) = Var\left[\frac{1}{N}\sum_{n=0}^{N}\Delta\mu_q^{(n)}\right] = \frac{Var\left[\Delta\mu_q^{(n)}\right]}{N}. \quad \text{(S15)}$$

Suppose we want the mean-squared error to be less than some value $\epsilon \ll 1$. How many samples ($N$) do we need to take to bring ourselves below this error on average? We have:

$$\frac{Var\left[\Delta\mu_q^{(n)}\right]}{N} < \epsilon \implies \frac{Var\left[\Delta\mu_q^{(n)}\right]}{\epsilon} < N. \quad \text{(S16)}$$

This means that increases in the variance of a weight estimate require proportionate increases in the number of samples required to reduce the error of the estimate. In practice, this requires high variance methods to process more data and to have lower learning rates, in some cases by several orders of magnitude. Even if a stochastic weight update is 'local' in a biologically-plausible sense, it may still require so much data for learning to occur as to be completely impractical.

## C.1 Comparing Variances

Analytic variance calculations are only possible for the simplest of examples, but the intuitions they provide are nevertheless valuable. In the sections that follow, we will show that samples from all three methods have exactly the same expectation (the 'signal'), but only IL and BP agree on their variance, while NVI* typically has much higher variance. For univariate normal distributions with identical variance, the loss $\mathcal{L} = \mathbb{E}_\lambda\left[KL(\tilde{q}||\tilde{p})\right] = KL[q||p] = T(\mu_p - \mu_q)^2/2\sigma^2$. Writing the variances in terms of the loss, we have:

$$Var_{\text{IL}} = Var_{\text{BP}} = \frac{T}{\sigma^2} \quad \text{(S17)}$$

$$Var_{\text{NVI}} = \frac{T}{2\sigma^2} + \frac{\mathcal{L}}{8\sigma^2}(3T+5) \quad \text{(S18)}$$

This shows that for the most part, IL and BP hugely outperform NVI*. However, it is possible for NVI* to outperform these methods in the limit as $\mathcal{L} \to 0$ (a regime only achieved *after* successful optimization). Here, as with our numerical results, we have incorporated two methods that partially ameliorate the high variance of the NVI* estimate, which for reasonably low-dimensional tasks, can still allow it to perform comparably to BP; however, NVI* is unlikely to scale well to high dimensions, even with these additions. The purpose for our analysis is to show that these high variance difficulties do not apply to IL, whose scaling properties are much closer to BP.

## C.2 Backpropagation

**Expectation** We will focus only on $\frac{d}{d\mu_q}$ for simplicity. Because the entropy of $\tilde{q}$ is constant with respect to the mean $\mu_q$, we don't have to worry about the second term in the objective function. Instead, we focus on:

$$-\frac{d}{d\mu_q}\mathcal{L} = \frac{d}{d\mu_q}\int\left[\int(\log\tilde{p}(r|\lambda))\tilde{q}(r|\lambda)dr\right]p(\lambda)d\lambda$$

$$= \frac{d}{d\mu_q}\sum_t\left[\int\frac{1}{2}(\log p(r_t))q(r_t)dr_t + \int\frac{1}{2}(\log q(r_t))p(r_t)dr_t\right]$$

$$= -\frac{d}{d\mu_q}\sum_t\left[\int\frac{1}{4\sigma^2}((r_t-\mu_p)^2)q(r_t)dr_t + \int\frac{1}{4\sigma^2}((r_t-\mu_q)^2)p(r_t)dr_t\right]. \quad \text{(S19)}$$

At this point, we employ the 'reparameterization trick,' which reduces the variance of the weight update relative to NVI*. For the first integral we use the change of variables $r_t = \mu_q + \eta_t$, and for the second integral we use the change of variables $r_t = \mu_p + \eta_t$, where $\eta_t \sim \mathcal{N}(0, \sigma^2)$. This gives:

$$-\frac{d}{d\mu_q}\mathcal{L} = -\frac{d}{d\mu_q}\sum_{t=0}^{T}\left[\int \frac{1}{4\sigma^2}((\mu_q + \eta_t - \mu_p)^2)p(\eta_t)d\eta_t + \int \frac{1}{4\sigma^2}((\mu_p + \eta_t - \mu_q)^2)p(\eta_t)d\eta_t\right]$$

$$= -\frac{d}{d\mu_q}\sum_{t=0}^{T}\int \frac{1}{2\sigma^2}((\mu_q + \eta_t - \mu_p)^2)p(\eta_t)d\eta_t$$

$$= \sum_{t=0}^{T}\int \frac{1}{\sigma^2}(\mu_p + \eta_t - \mu_q)p(\eta_t)d\eta_t. \tag{S20}$$

Computing this expectation analytically, we have: $-\frac{d}{d\mu_q}\mathcal{L} = \frac{T}{\sigma^2}(\mu_p - \mu_q)$, which is unbiased, because we have not employed any approximations. If we were to compute this expectation using samples from $p(\eta_t)$, each individual parameter update would be given by $\Delta\mu_q \propto \sum_{t=0}^{T}\frac{1}{\sigma^2}(\mu_p + \eta_t - \mu_q)$ for a given sample from $\eta$. Given our expected weight update, we now ask for the variance.

**Variance**  The variance of a sample, $\sum_{t=0}^{T}\frac{1}{\sigma^2}(\mu_p + \eta_t - \mu_q)$, is given by:

$$Var(\Delta\mu_q) = \int \left(\frac{1}{\sigma^2}(\sum_{t=0}^{T}(\mu_p + \eta_t - \mu_q - (\mu_p - \mu_q)))\right)^2 p(\eta_t)d\eta_t$$

$$= \int \sum_{t=0}^{T}\frac{\eta_t^2}{\sigma^4}p(\eta_t)d\eta_t$$

$$= \frac{T}{\sigma^2}. \tag{S21}$$

### C.3   Impression learning

**Expectation**  We can use our previous derivation of the IL weight update to write:

$$-\frac{d}{d\mu_q}\mathcal{L} \approx 2\sum_{t=0}^{T}\left[\int \left[(1 - \lambda_t)\frac{d}{d\mu_q}\log q(r_t) + (\lambda_t)\frac{d}{d\mu_q}\log p\right]\tilde{q}(r_t|\lambda_t)dr_t\right]p(\lambda_t)d\lambda_t$$

$$= 2\sum_{t=0}^{T}\left[\int (1 - \lambda_t)\frac{d}{d\mu_q}\log q(r_t)]\tilde{q}(r_t|\lambda)dr_t\right]p(\lambda_t)d\lambda_t$$

$$= \sum_{t=0}^{T}\int \frac{d}{d\mu_q}\log q(r_t)p(r_t)dr_t \tag{S22}$$

where this last equality follows from the fact that $\tilde{q}(r_t|\lambda) = p(r_t)$ whenever $1 - \lambda_t = 1$. Continuing our derivation by substituting in $\log q(r_t)$ and discarding constants, we have:

$$-\frac{d}{d\mu_q}\mathcal{L} \approx \sum_{t=0}^{T}\int -\frac{d}{d\mu_q}\frac{1}{2\sigma^2}(r_t - \mu_q)^2 p(r_t)dr_t$$

$$= \sum_{t=0}^{T}\int \frac{1}{\sigma^2}(r_t - \mu_q)p(r_t)dr_t. \tag{S23}$$

Computing this expectation analytically gives: $-\frac{d}{d\mu_q}\mathcal{L} \approx \frac{T}{\sigma^2}(\mu_p - \mu_q)$. Interestingly, in this case, the expected weight update coincides directly with the update given by BP, meaning that for this contrived example, IL is unbiased. This is clearly not the case in general, but works because our simplified network has no temporal interdependencies between variables and lacks hierarchical structure. In fact, the IL update also directly corresponds to the WS update in this case for the same reason. As with BP, we can ask about the variance of an individual sample of an update given by IL, assuming $\Delta\mu_q \propto \sum_{t=0}^{T}\frac{1}{\sigma^2}(r_t - \mu_q)$.

**Variance**   The variance of a sample, $\sum_{t=0}^{T} \frac{1}{\sigma^2}(r_t - \mu_q)$, is given by:

$$Var(\Delta\mu_q) = \int \left( \frac{1}{\sigma^2} (\sum_{t=0}^{T} r_t - \mu_q - (\mu_p - \mu_q)) \right)^2 p(r_t)dr_t$$

$$= \int \frac{1}{\sigma^4}(\sum_{t=0}^{T}(r_t - \mu_p))^2 p(r_t)dr_t$$

$$= \int \frac{1}{\sigma^4} \sum_{t=0}^{T} \sum_{t'=0}^{T}(r_t - \mu_p)(r_{t'} - \mu_p)p(r_t)dr_t$$

$$= \int \frac{1}{\sigma^4} \sum_{t=0}^{T}(r_t - \mu_p)^2 p(r_t)dr_t$$

$$= \frac{T}{\sigma^2}, \tag{S24}$$

where here we have exploited the fact that $\mathbb{E}[(r_t - \mu_p)(r_{t'} - \mu_p)] = 0 \ \forall t \neq t'$. This shows that for this simple example, there is a perfect correspondence between both the expectation and the variance of IL compared to BP.

### C.4   Neural Variational Inference

**Expectation**   The difference between NVI* and BP is that we do not use a change of variables. Given our previous derivation of the NVI* update (Eq. S4), we have:

$$-\frac{d}{d\mu_q}\mathcal{L} = \int \left[ \int \frac{d}{d\mu_q} \log \tilde{p}(r|\lambda_t)\tilde{q}(r|\lambda) + (\log \tilde{p} - \log \tilde{q})(\frac{d}{d\mu_q} \log \tilde{q}(r|\lambda))\tilde{q}(r|\lambda)dr \right] p(\lambda_t)d\lambda_t$$

$$= \int \left[ \int \left( \sum_{t=0}^{T} \frac{(1-\lambda_t)}{\sigma^2}(r_t - \mu_q) + (\log \tilde{p} - \log \tilde{q})\sum_{t=0}^{T} \frac{\lambda_t}{\sigma^2}(r_t - \mu_q) \right) \tilde{q}(r|\lambda)dr \right] p(\lambda_t)d\lambda_t,$$

where the second equality follows from substituting in $\frac{d}{d\mu_q} \log \tilde{p}(r|\lambda_t)$ and $\frac{d}{d\mu_q} \log \tilde{q}(r|\lambda)$. Noting that $\log \tilde{p} - \log \tilde{q} = \log p - \log q$ when $\lambda_t = 1$, we continue:

$$-\frac{d}{d\mu_q}\mathcal{L} = \int \left[ \int \left( \sum_{t=0}^{T} \frac{(1-\lambda_t)}{\sigma^2}(r_t - \mu_q) + (\log p - \log q)\sum_{t=0}^{T} \frac{\lambda_t}{\sigma^2}(r_t - \mu_q) \right) \tilde{q}(r|\lambda)dr \right] p(\lambda_t)d\lambda_t$$

$$= \mathbb{E}_{r,\lambda} \left[ \sum_{t=0}^{T} \frac{(1-\lambda_t)}{\sigma^2}(r_t - \mu_q) - \left( \sum_{t=0}^{T}(r_t - \mu_p)^2 - (r_t - \mu_q)^2 \right) \sum_{t=0}^{T} \frac{\lambda_t}{2\sigma^4}(r_t - \mu_q) \right]$$

$$= \mathbb{E}_{r,\lambda} \left[ \sum_{t=0}^{T} \frac{(1-\lambda_t)}{\sigma^2}(r_t - \mu_q) - \left( \sum_{t=0}^{T} 2r_t(\mu_q - \mu_p) + \mu_p^2 - \mu_q^2 \right) \sum_{t=0}^{T} \frac{\lambda_t}{2\sigma^4}(r_t - \mu_q) \right]. \tag{S25}$$

At this point, we'll allow ourselves to exploit the structure of our problem in two ways commonly employed in NVI*. First, we observe that the loss at a particular time step, $2r_t(\mu_q - \mu_p) + \mu_p^2 - \mu_q^2$ is independent of $r_{t'} - \mu_q$ for $t' > t$, i.e. fluctuations in variables at future time steps do not influence the current loss. Incorporating this fact modifies our update to give:

$$-\frac{d}{d\mu_q}\mathcal{L} = \mathbb{E}_{r,\lambda} \left[ \sum_{t=0}^{T} \frac{(1-\lambda_t)}{\sigma^2}(r_t - \mu_q) - \sum_{t=0}^{T}\sum_{t'\leq t} \frac{\lambda_t}{2\sigma^4} \left( 2r_t(\mu_q - \mu_p) + \mu_p^2 - \mu_q^2 \right) (r_{t'}' - \mu_q) \right]. \tag{S26}$$

Next, we notice that $\mathbb{E}\left[\sum_{t' \leq t} \frac{\lambda_t}{2\sigma^4}(r'_t - \mu_q)\right] = 0$, so we can subtract from our update $a \times \sum_{t' \leq t} \frac{\lambda_t}{2\sigma^4}(r'_t - \mu_q)$ for some constant $a$, without modifying the expectation of our loss. Choosing a constant $a$ that will reduce the variance of the parameter update is a common technique used in NVI*, called using a 'control variate' [1, 2]. We notice that the average loss contributes nothing to the expectation, so we take $a = 2\mu_q(\mu_q - \mu_p) + \mu_p^2 - \mu_q^2$, giving the improved-variance update:

$$-\frac{d}{d\mu_q}\mathcal{L} = \mathbb{E}_{r,\lambda}\left[\sum_{t=0}^{T}\frac{(1-\lambda_t)}{\sigma^2}(r_t - \mu_q) - \sum_{t=0}^{T}\sum_{t' \leq t}\frac{\lambda_t}{\sigma^4}(r_t - \mu_q)(\mu_q - \mu_p)(r'_t - \mu_q)\right]. \quad \text{(S27)}$$

Individual samples from this method of differentiation are more complicated (and higher variance) than IL or BP. An individual sample would give: $\sum_{t=0}^{T}\frac{(1-\lambda_t)}{\sigma^2}(r_t - \mu_q) - \sum_{t=0}^{T}\sum_{t' \leq t}\frac{\lambda_t}{\sigma^4}(r_t - \mu_q)(\mu_q - \mu_p)(r'_t - \mu_q)$. We'll first compute the expectation of this expression (to verify that it is equivalent to BP and IL), and then we'll compute its variance. Continuing our calculation, we get:

$$
\begin{aligned}
-\frac{d}{d\mu_q}\mathcal{L} &= \mathbb{E}_{r,\lambda}\left[\sum_{t=0}^{T}\frac{1-\lambda_t}{\sigma^2}(r_t - \mu_q) - \sum_{t=0}^{T}\sum_{t' \leq t}\frac{\lambda_t}{\sigma^4}(r_t - \mu_q)(\mu_q - \mu_p)(r'_t - \mu_q)\right] \\
&= \int \sum_{t=0}^{T}\frac{(1-\lambda_t)}{\sigma^2}(r_t - \mu_q)p(r)dr + \int \frac{1}{2\sigma^4}\sum_{t=0}^{T}\sum_{t' \leq t}(r_t - \mu_q)(\mu_p - \mu_q)(r'_t - \mu_q)q(r)dr \\
&= \frac{T}{2\sigma^2}(\mu_p - \mu_q) + \int \frac{(\mu_p - \mu_q)}{2\sigma^4}\sum_{t=0}^{T}\sum_{t' \leq t}(r_t - \mu_q)(r'_t - \mu_q)q(r)dr \\
&= \frac{T}{2\sigma^2}(\mu_p - \mu_q) + \int \frac{(\mu_p - \mu_q)}{2\sigma^4}\sum_{t=0}^{T}\sum_{t' \leq t}(\eta_t)(\eta_{t'})p(\eta)d\eta \\
&= \frac{T}{2\sigma^2}(\mu_p - \mu_q) + \int \frac{(\mu_p - \mu_q)}{2\sigma^4}\sum_{t=0}^{T}\eta_t^2 p(\eta)d\eta \\
&= \frac{T}{\sigma^2}(\mu_p - \mu_q), \quad \text{(S28)}
\end{aligned}
$$

where the fourth equality comes from reparameterizing with the transformation $\eta_t = r_t - \mu_q$ and the fifth equality stems from the fact that $\mathbb{E}[\eta_t] = 0$ and $\mathbb{E}[\eta_t \eta_{t'}] = 0$. This verifies that whether we sample over $r$ using the black-box differentiation method, or over $\eta$ using the reparameterization trick, or use IL, we will arrive at the same weight update in *expectation*. The variance of sample estimates thus distinguishes IL from NVI* (on this example at least).

**Variance**   Because of the NVI* sample estimate's increased complexity, the variance calculation is also much more involved:

$$
\begin{aligned}
Var(\Delta\mu_q) &= \mathbb{E}_{r,\lambda}\left[\left(\Delta\mu_q - \frac{T}{\sigma^2}(\mu_p - \mu_q)\right)^2\right] \\
&= \mathbb{E}_{r,\lambda}\left[\left(\sum_{t=0}^{T}\frac{(1-\lambda_t)}{\sigma^2}(r_t - \mu_q) - \sum_{t=0}^{T}\sum_{t' \leq t}\frac{\lambda_t}{2\sigma^4}(r_t - \mu_q)(\mu_q - \mu_p)(r'_t - \mu_q) - \frac{T}{\sigma^2}(\mu_p - \mu_q)\right)^2\right] \\
&= \frac{1}{2}\int \frac{1}{\sigma^4}\sum_{t=0}^{T}(r_t - \mu_p)^2 p(r)dr \\
&\quad + \frac{1}{2}\int \left(\frac{1}{2\sigma^4}\sum_{t=0}^{T}\sum_{t' \leq t}(r_t - \mu_q)(\mu_p - \mu_q)(r'_t - \mu_q) - \frac{T}{\sigma^2}(\mu_p - \mu_q)\right)^2 q(r)dr, \\
& \quad \text{(S29)}
\end{aligned}
$$

where in this last step we have taken an expectation over $\lambda$, observing that the first term is only nonzero if $\lambda_t = 0$, and the second term is only nonzero if $\lambda_t = 1$. Now we apply the reparameterization, taking $r_t = \eta_t + \mu_p$ in the first integral, and $r_t = \eta_t + \mu_q$ in the second integral, giving:

$$
\begin{aligned}
Var(\Delta\mu_q) =& \frac{T}{2\sigma^2} + \frac{1}{2} \int \left( \frac{1}{2\sigma^4} \sum_{t=0}^{T} \sum_{t' \leq t} \left( \eta_t (\mu_p - \mu_q) \right) (\eta_{t'}) - \frac{T}{\sigma^2} (\mu_p - \mu_q) \right)^2 p(\eta) d\eta \\
=& \frac{T}{2\sigma^2} + \frac{(\mu_p - \mu_q)^2}{2\sigma^4} \int \left( \frac{1}{2\sigma^2} \sum_{t=0}^{T} \sum_{t' \leq t} \eta_t \eta_{t'} - T \right)^2 p(\eta) d\eta \\
=& \frac{T}{2\sigma^2} + \frac{(\mu_p - \mu_q)^2}{2\sigma^4} \mathbb{E}_{\eta_t} \left[ \left( \frac{1}{2\sigma^2} \sum_{t=0}^{T} \sum_{t' \leq t} \eta_t \eta_{t'} \right)^2 - \frac{T}{\sigma^2} \left( \sum_{t=0}^{T} \sum_{t' \leq t} \eta_t \eta_{t'} \right) + T^2 \right] \\
=& \frac{T}{2\sigma^2} + \frac{(\mu_p - \mu_q)^2}{2\sigma^4} \mathbb{E}_{\eta_t} \left[ \left( \frac{1}{2\sigma^2} \sum_{t=0}^{T} \sum_{t' \leq t} \eta_t \eta_{t'} \right)^2 - \frac{T}{\sigma^2} \left( \sum_{t=0}^{T} \eta_t^2 \right) + T^2 \right] \\
=& \frac{T}{2\sigma^2} + \frac{(\mu_p - \mu_q)^2}{2\sigma^4} \mathbb{E}_{\eta_t} \left[ \left( \frac{1}{2\sigma^2} \sum_{t=0}^{T} \sum_{t' \leq t} \eta_t \eta_{t'} \right)^2 \right] \\
=& \frac{T}{2\sigma^2} + \frac{(\mu_p - \mu_q)^2}{8\sigma^8} \mathbb{E}_{\eta_t} \left[ \sum_{t=0}^{T} \sum_{t'=0}^{T} \sum_{t'' \leq t} \sum_{t''' \leq t'} \eta_t \eta_{t'} \eta_{t''} \eta_{t'''} \right] \\
=& \frac{T}{2\sigma^2} + \frac{(\mu_p - \mu_q)^2}{8\sigma^8} \sum_{t=0}^{T} \sum_{t'=0}^{T} \sum_{t'' \leq t} \sum_{t''' \leq t'} \mathbb{E}_{\eta_t} \left[ \eta_t \eta_{t'} \eta_{t''} \eta_{t'''} \right]. \quad (S30)
\end{aligned}
$$

Now, we notice that there are three mutually exclusive and exhaustive conditions under which this expectation is nonzero, using the the fact that only the even moments of the normal distribution are nonzero:

$$
\mathbb{E}_{\eta_t} \left[ \eta_t \eta_{t'} \eta_{t''} \eta_{t'''} \right] = \begin{cases} \sigma^4 & \text{if } t = t' \text{ and } t'' = t''' \text{ and } t \neq t'' \\ \sigma^4 & \text{if } t = t'' \text{ and } t' = t''' \text{ and } t \neq t' \\ 3\sigma^4 & \text{if } t = t' = t'' = t''' \\ 0 & \text{otherwise.} \end{cases} \quad (S31)
$$

These three different conditions result in three different sums:

$$
\begin{aligned}
Var(\Delta\mu_q) =& \frac{T}{2\sigma^2} + \frac{(\mu_p - \mu_q)^2}{8\sigma^8} \left( \sum_{t=1}^{T} \sum_{t' < t} \sigma^4 + \sum_{t=0}^{T} \sum_{t' \neq t} \sigma^4 + \sum_{t=0}^{T} 3\sigma^4 \right) \\
=& \frac{T}{2\sigma^2} + \frac{(\mu_p - \mu_q)^2}{8\sigma^8} \left( \sigma^4 \sum_{t=1}^{T} (t) + T(T-1)\sigma^4 + 3T\sigma^4 \right) \\
=& \frac{T}{2\sigma^2} + \frac{(\mu_p - \mu_q)^2}{8\sigma^8} \left( \frac{1}{2}T(T+1)\sigma^4 + T(T-1)\sigma^4 + 3T\sigma^4 \right) \\
=& \frac{T}{2\sigma^2} + \frac{(\mu_p - \mu_q)^2}{16\sigma^4} \left( 3T^2 + 5T \right) \\
=& \frac{T}{2\sigma^2} + \frac{\mathcal{L}}{8\sigma^2} \left( 3T + 5 \right), \quad (S32)
\end{aligned}
$$

where the third equality follows from the arithmetic series identity: $\sum_{t=1}^{T}(t) = \frac{1}{2}T(T+1)$.

# D  Multilayer Network Architecture

Here we outline the architecture for the 2-layer network used for processing the Free Spoken Digits dataset [15] in Figure 4.

## D.1  Model structure

Our inference architecture simply adds an additional feedforward layer of neurons to the network:

$$\mathbf{s}_t^{\text{inf}} = \mathbf{z}_t + \sigma_s^{\text{inf}}\boldsymbol{\xi}_t \tag{S33}$$

$$\mathbf{r}_t^{\text{inf1}} = f(\mathbf{W}_1\mathbf{s}_t + \mathbf{a}) + \sigma_1^{\text{inf}}\boldsymbol{\eta}_t^1 \tag{S34}$$

$$\mathbf{r}_t^{\text{inf2}} = f(\mathbf{W}_2\mathbf{r}_t^{\text{inf1}}) + \sigma_2^{\text{inf}}\boldsymbol{\eta}_t^2, \tag{S35}$$

where $\mathbf{W}_l$ denotes the feedforward weights from layer $l-1$ to layer $l$, $\mathbf{a}$ is an additive bias parameter, $\boldsymbol{\eta}_t^1, \boldsymbol{\eta}_t^2, \boldsymbol{\xi}_t \sim \mathcal{N}(0,1)$ are independent white noise samples, $\sigma_1^{\text{inf}}, \sigma_2^{\text{inf}}$, and $\sigma_s^{\text{inf}}$ denote the inference standard deviations for their respective layers, and the nonlinearity $f(\cdot)$ is the $\tanh$ function. The multilayer generative model includes an additional feedforward decoder step:

$$\mathbf{r}_t^{\text{gen2}} = ((1-k_t)\mathbf{D}_2 + k_t\mathbf{I})\, r_{t-1} + \sigma_2^{\text{gen}}\boldsymbol{\eta}_t^2 \tag{S36}$$

$$\mathbf{r}_t^{\text{gen1}} = f(\mathbf{D}_1\mathbf{r}_t^{\text{gen2}} + \mathbf{b}) + \sigma_1^{\text{gen}}\boldsymbol{\eta}_t^1 \tag{S37}$$

$$\mathbf{s}_t^{\text{gen}} = f(\mathbf{D}_s\mathbf{r}_t^{\text{gen1}}) + \sigma_s^{\text{gen}}\boldsymbol{\xi}_t, \tag{S38}$$

where $\mathbf{D}_2$ is a diagonal transition matrix, $\mathbf{D}_1$ and $\mathbf{D}_s$ are prediction weights to their layers from higher layers, $\mathbf{b}$ is an additive bias parameter, $\mathbf{I}$ is the identity matrix, and $\sigma_1^{\text{gen}}, \sigma_2^{\text{gen}}$, and $\sigma_s^{\text{gen}}$ denote the generative standard deviations for their layers. We define $k_t$ as in the 1-layer network. Also in keeping with the basic model, during simulation, samples are determined by a combination of $p_m$ and $q$, given by $\tilde{q}_\theta$:

$$\mathbf{r}_t^2 = \lambda_t\mathbf{r}_t^{\text{inf2}} + (1-\lambda_t)\mathbf{r}_t^{\text{gen2}} \tag{S39}$$

$$\mathbf{r}_t^1 = \lambda_t\mathbf{r}_t^{\text{inf1}} + (1-\lambda_t)\mathbf{r}_t^{\text{gen1}} \tag{S40}$$

$$\mathbf{s}_t = \lambda_t\mathbf{s}_t^{\text{inf}} + (1-\lambda_t)\mathbf{s}_t^{\text{gen}}. \tag{S41}$$

## D.2  Parameter updates

Adding additional layers to our model does not change the fact that the parameter updates can be interpreted as local synaptic plasticity rules at the basal (for $q$) or apical (for $p$) compartments of our neuron model. Plugging our probability models into the equation for the IL parameter update (Eq. 5), calculating derivatives, and updating our parameters stochastically at every time step as with our basic model gives:

$$\Delta\mathbf{W}_1^{(ij)} \propto \frac{1-\lambda_t}{(\sigma_1^{\text{inf}})^2}((\mathbf{r}_t^1)^{(i)} - f(\mathbf{W}_1\mathbf{s}_t + \mathbf{a})^{(i)})f'(\mathbf{W}_1\mathbf{s}_t + \mathbf{a})^{(i)}\mathbf{s}_t^{(j)} \tag{S42}$$

$$\Delta\mathbf{a}^{(i)} \propto \frac{1-\lambda_t}{(\sigma_1^{\text{inf}})^2}((\mathbf{r}_t^1)^{(i)} - f(\mathbf{W}_1\mathbf{s}_t + \mathbf{a})^{(i)})f'(\mathbf{W}_1\mathbf{s}_t + \mathbf{a})^{(i)} \tag{S43}$$

$$\Delta\mathbf{W}_2^{(ij)} \propto \frac{1-\lambda_t}{(\sigma_2^{\text{inf}})^2}((\mathbf{r}_t^2)^{(i)} - f(\mathbf{W}_2\mathbf{r}_t^1)^{(i)})f'(\mathbf{W}_2\mathbf{r}_t^1)^{(i)}(\mathbf{r}_t^1)^{(j)} \tag{S44}$$

$$\Delta\mathbf{D}_2^{(ii)} \propto \frac{\lambda_t(1-k_t)}{(\sigma_2^{\text{gen}})^2}((\mathbf{r}_t^2)^{(i)} - (\mathbf{D}_2\mathbf{r}_{t-1}^2)^{(i)})(\mathbf{r}_{t-1}^2)^{(i)} \tag{S45}$$

$$\Delta\mathbf{D}_1^{(ij)} \propto \frac{\lambda_t}{(\sigma_s^{\text{gen}})^2}((\mathbf{r}_t^1)^{(i)} - f(\mathbf{D}_1\mathbf{r}_t^2 + \mathbf{b})^{(i)})f'(\mathbf{D}_1\mathbf{r}_t^2 + \mathbf{b})^{(i)}(\mathbf{r}_t^2)^{(j)} \tag{S46}$$

$$\Delta\mathbf{b}^{(i)} \propto \frac{\lambda_t}{(\sigma_s^{\text{gen}})^2}((\mathbf{r}_t^1)^{(i)} - f(\mathbf{D}_1\mathbf{r}_t^2 + \mathbf{b})^{(i)})f'(\mathbf{D}_1\mathbf{r}_t^2 + \mathbf{b})^{(i)} \tag{S47}$$

$$\Delta\mathbf{D}_s^{(ij)} \propto \frac{\lambda_t}{(\sigma_s^{\text{gen}})^2}(\mathbf{s}_t^i - f(\mathbf{D}_s\mathbf{r}_t^1)^{(i)})f'(\mathbf{D}_s\mathbf{r}_t^1)^{(i)}(\mathbf{r}_t^1)^{(j)}. \tag{S48}$$

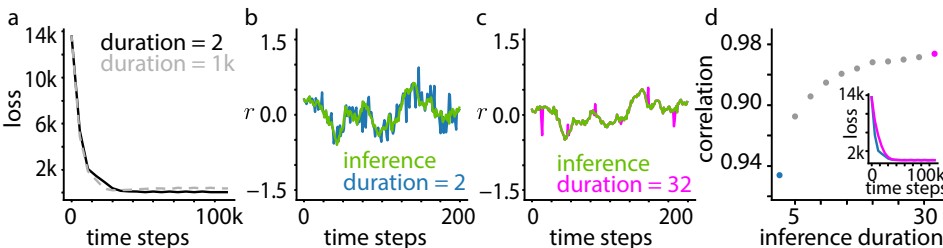

Figure S1: **Additional variations on the phase duration. a.** Comparison of ELBO loss for IL (black) to WS with a 1000 time step phase duration (gray) over training. **b.** Comparison of an example neuron's activity through time when the network is in inference mode (green, $\lambda_t = 1$) and when the network is alternating phase, spending 2 time steps in the inference phase, and two time steps the generative phase (blue); the random seed and stimuli are identical in both cases. **c.** Same as b, but the alternating network spends 32 time steps in the inference phase. **d.** The correlation across time between neurons in inference mode vs. while alternating phase, for identical random seeds. The inference duration is incremented, while the generative duration is kept constant at 2 time steps. Inset shows the loss for an inference duration of 2 (blue) compared to the loss for an inference duration of 32 (pink).