# OpenReview forum: "Impression learning: Online representation learning with synaptic plasticity"
_NeurIPS.cc/2021/Conference — NeurIPS 2021 Poster_

### Official Review · Reviewer_gqEo · 2021-07-11

**Rating:** 8
**Confidence:** 2

**Summary:**

The authors are interested in understanding how brain circuits are able to perform unsupervised learning. The work is an extension of the wake-sleep algorithm, where the algorithm is asked to perform two functions: i) infer latent variables from observations and ii) build a generative model of the received observations. They start by defining a loss function to be minimized for their system to perform both functions and then interpret this minimization as a plasticity rule between the weights of a network of neurons. Their learning algorithm is compared to other existing one in the literature, and its efficiency is probed on a real-world task.

**Limitations And Societal Impact:**

Yes

**Main Review:**

Although I am not an expert on this topic I have enjoyed reading the paper. The problem of designing unsupervised learning algorithm that could be mapped onto the functioning of brain circuits is of fundamental importance for both neuroscience and AI, and this work seems a solid contribution towards this goal. The paper is well written and clearly situates its contribution with respect to the existing literature. The direct comparison with other existing approaches, as well as the probing of performance of the algorithm on a real-world task make the work appears as high-quality. The mapping between the inference part of the algorithm and processing at the basal part of neurons, and the mapping between the generative part of the algorithm and processing at the apical part of neurons seem well sounded and allows to discuss experimental results in neuroscience. It would be nice to mention how current views on plasticity relates to the idea that weight updates should depend from firing rates of pre- and post- synaptic cells as well as basal current.




**Time Spent Reviewing:**

5

---

> ### Author Response · Authors · 2021-08-09
> **Review Response (gqEo)**
>
> Q. It would be nice to mention how current views on plasticity relates to the idea that weight updates should depend from firing rates of pre- and post- synaptic cells as well as basal current.
>
> A. There are several individual experimental results to show that basal plasticity is influenced by apical dendritic calcium spikes (Bittner et. al, 2015) as well as results showing that backpropagating action potentials from the soma can trigger plasticity in distal dendrites (Magee & Johnston, 1997), but to our knowledge the exact form of the plasticity rule (prediction error) has not yet been confirmed. We will elaborate on the relationships between our model and existing theoretical and experimental work. In particular, several previous theories (e.g. Urbanzik & Senn 2014; Payeur & Naud 2020) have postulated similar interactions between apical and basal compartments, but they do not apply to online learning of temporal statistics.

---

### Official Review · Reviewer_LAYW · 2021-07-15

**Rating:** 6
**Confidence:** 3

**Summary:**

In this paper the authors introduce a framework for approximate online Bayesian inference in an unsupervised setting. This framework generalises the wake-sleep algorithm and enables the derivation of a local learning rule. The generative and inference components of the model are mapped onto distal and basal dendrites, respectively. A number of experiments are conducted to demonstrate the model behaviour using both artificial (toy) and naturalistic tasks.

**Limitations And Societal Impact:**

There are a number of limitations that have not been fully addressed in terms of relating the model to biology (see comments above).

**Main Review:**

The framework introduced by the authors is interesting and novel, and it provides a generalization of the wake sleep algorithm. The paper is well written, but some elements are missing to make the contribution clear (see below). The main contribution is the derivation of a local learning rule with elements that can be mapped onto distal and basal dendrites. This could be of high significance for computational neuroscience but its at present hard to fully assess (see below).

*Major points*:
 - The link between the learning rule derived and biology is far from clear. There is only a very brief discussion on this in page 5 and the Discussion. But if this model is to be considered for understanding how basal/distal interactions lead to synaptic plasticity, this needs more discussion.
 - Importantly the basal learning rule (14) does not seem to depend on distal activity? This would be at odds with biology, as there is strong evidence that distal activity drives basal plasticity (see for example work by the labs of Anthony Holtmaat, Jeffrey Magee and Jesper Sjostrom).
 - Given that the algorithm proposed can do faster/online learning compared to the WS algorithm, why was it not used as one of the comparisons? It seems like one of the key models to contrast with. In particular, it would be important to show that this model does indeed need shorter offline phases. This seems to be one of the key contributions here that does not appear to have been clearly demonstrated.
 - In general a closer link between the model and existing/recent observations in how basal/distal interact would strength the story.


*Minor points*:

 - 119: What is the biological interpretation for the gating variable, k?

 - 160: Does the network still learn if the latent space dimension or generative noise level do not match their ground truth values?

 - 226: It would be interesting to mention if the network can learn when the phase duration is not the same for the two phases or if it can vary over the course of training.

 - 274: I think the distinction between excitatory and inhibitory neurons is an important point. Perhaps this could be elaborated more.

 - Fig 2c, 2d, 3b: choice of colours for NVI vs IL could be more distinctive

- The authors use D donation for both Diagonal and non-diagonal matrices. This can cause confusion, in particular in Appendix D.1 (model structure), where D_2 is the diagonal transition matrix, and D_1 is the prediction weights. I suggest authors use different matrix notation or more informative subscripts.

 - In Eq.6, when the /lambda switches to 0 from 1, the value of k becomes 1, which in result the r^{gen}_t = r_{t-1} + \sigma_{r}^{gen} \eta_t . However, in Eq.15, there is no update when k_t = 1. So, it is not clear why this gating bias is in the training of the generative transition parameters D_r when there is no update at that time step. Can you please clarify this.

 - In Eq. 3, the second line, [log /hat{p} / \hat{q}] are approximated by Taylor expansion around 1. It is not clear why it is a valid expansion about 1. The authors explain in detail how this leads to introducing bias in parameter update in appendix A, but it would be good to briefly explain what are the consequences of this approximation, in particular as this seems to be a novel element compared to previous work.



**Time Spent Reviewing:**

5

---

> ### Author Response · Authors · 2021-08-09
> **Review Response (LAYW)**
>
> Q. Importantly the basal learning rule (14) does not seem to depend on distal activity?
>
> A. It is somewhat subtle, but the basal learning rule does depend on distal activity. For an update to occur, the network must be in the generative phase (lambda_t = 0), which means that the firing rates of neurons are determined by the apical synapses rather than the basal synapses (Eq. 14). This means that the firing rate variable r_t is determined by input to the apical synapses, not by input to the basal synapses. We’ll add this point to the text.
>
> Q. Given that the algorithm proposed can do faster/online learning compared to the WS algorithm, why was it not used as one of the comparisons?
>
> A. Figure 3f. provides a comparison between impression learning for a short phase duration (2) and a long phase duration (32). As we note on lines 224-25 and in Appendix B.3, the long phase duration case can be thought of as an online version of Wake/Sleep. In this case, the network is spending lengthy periods of time training its apical synapses, and lengthy periods of time training its basal synapses, instead of training them all together. These two approaches perform similarly, but Wake/Sleep requires a lengthy ‘sleep’ training period, while impression learning allows the system to train while receiving input, without significantly disrupting normal neural activity (Fig. 3e)--this is the critical difference between the two algorithms. We agree that a comparison to the offline Wake/Sleep algorithm would be valuable, and we will provide one in a supplementary figure.
>
> Q. In general a closer link between the model and existing/recent observations in how basal/distal interact would strength the story.
>
> A. We will add a more detailed comparison to the recent experimental literature. In addition to work on the link between distal stimulation and basal synaptic plasticity (Holtmaat, Magee and Sjostrom), we will discuss two new experimental studies that have analyzed the different tuning properties of apical and basal dendritic compartments in pyramidal neurons over the course of learning (Gillon & Richards 2020; Rashid & Basu 2021).
>
> Q. 119: What is the biological interpretation for the gating variable, k?
>
> A. As discussed in Lines 120-6, k_t essentially enforces that parameters are not updated for a short period of time after the phase switches from ‘generative’ to ‘inference’ (Eq. 15). This could be implemented with a slight delay in activating learning for the apical parameters, caused by a cascade of biochemical processes similar to those proposed in the context of synaptic plasticity theories (Fusi et. al, 2005; Clopath et. al 2008).
>
> Q. 160: Does the network still learn if the latent space dimension or generative noise level do not match their ground truth values?
>
> A. Our neural network model is guaranteed to have a mismatch in latent space dimension and generative noise levels compared to the true latent structure of the Free Spoken Digits Dataset. In this context, the network is still able to learn the task, so in this case, model mismatch does not prevent learning. But in general, it is possible that the unlearned parameters or structural restrictions of either the inference or generative models will render a good fit impossible.
>
> Q. 226: It would be interesting to mention if the network can learn when the phase duration is not the same for the two phases or if it can vary over the course of training.
>
> A. Our mathematical derivation relies on an assumption that for any sequence lambda_t, the opposite sequence (1-lambda_t) is equally probable (Lines 101-2). However, empirically we found that training under asymmetric conditions (long inference phase, short generative phase) with the same plasticity rule still produces good learning (to be added in Suppl.).
>
> Q. 274: I think the distinction between excitatory and inhibitory neurons is an important point. Perhaps this could be elaborated more.
>
> A. Distinguishing between excitatory and inhibitory neurons in our network essentially amounts to enforcing Dale’s Law, which was once considered a fundamental property of neural systems (these days, more controversial). Our model can handle this feature, in contrast to many alternative local learning models which require symmetric recurrent connectivity (e.g. Hopfield networks, Boltzmann machines). We will elaborate on this point in the text.
>
> Q. In Eq.6, when the /lambda switches to 0 from 1, the value of k becomes 1, which in result the r^{gen}t = r{t-1} + \sigma_{r}^{gen} \eta_t . However, in Eq.15, there is no update when k_t = 1. So, it is not clear why this gating bias is in the training of the generative transition parameters D_r when there is no update at that time step.
>
> A. We include k_t as a gating variable because the statistics of r_t given r_{t-1} are different when a switch has just occurred, compared to when the network has been in ‘inference’ mode (lambda_t = 1) for at least one time step (see Line 120-123). To prevent the parameter update for D_r being biased by this difference in statistics, we designed our generative model to have a different parametric form directly after a switch has occurred. Essentially, (1-k_t) appears in the update to indicate that the generative parameters should not be updated immediately after lambda_t has switched from 0 to 1, because the parameters do not influence r^gen_t at that time step.
>
> Q. In Eq. 3, the second line, [log /hat{p} / \hat{q}] are approximated by Taylor expansion around 1. It would be good to briefly explain what are the consequences of this approximation.
>
> A. If our objective were perfectly optimal, we would have \hat{p} = \hat{q}, so \hat{p} / \hat{q} = 1, and log (\hat{p} / \hat{q}) = 0. Effectively, we are Taylor expanding the gradient update around the global optimum, which helps ensure that the parameter update, despite being an approximation, behaves properly at the global optimum. The further the network state is from the global optimum, and the greater the mismatch between the model and the inference/ground truth stimuli, the less accurate this update will be. This is why we need to provide empirical validation that our parameter update aligns well with the true gradient over learning (Fig. 2b).
> It may also help to note that the Wake/Sleep algorithm implicitly performs this approximation (because it is a special case of our framework, see appendix B.3).

---

> > ### Comment · Reviewer_LAYW · 2021-08-11
> > **Reply to authors**
> >
> > Thank you for the detailed reply. I am happy with the answers provided, but will maintain my score as the biological plausibility and link to experimental observations is still not clear to me.

---

### Official Review · Reviewer_YoDe · 2021-07-16

**Rating:** 7
**Confidence:** 4

**Summary:**

The authors proposed an algorithm called impression learning as a mechanism for the brain to learn the generative model of sensory inputs and meanwhile performs probabilistic inference of this model. The authors proposed a local synaptic learning rule to learn the generative model, and considered sampling-based inference. It is a valuable contribution to our understanding the general principle of neural coding.

**Limitations And Societal Impact:**

Not applicable.

**Main Review:**

Originality:
The main idea is based on wake-sleep algorithm, but has novel non-trivial improvements.
Strength:
This work proposes a novel mechanism for the brain to perform approximate online Bayesian inference through a local learning rule, which is quite interesting.
Weaknesses:
The biological plausibility of some elements need to be clarified.
1) what is the biological meaning of D_r in equation 6 ?  D_r  seems to be a recurrent connection matrix in the latent layer, but it is  diagonal and does not play a role when k_t=1.
2) the neuron should receive both top-down (apical) and bottom-up (basal) inputs concurrently，the variable λ_t induced by dopamine may control the learning phase (the infer phase and generative phase), but should not also control the apical inputs and basal inputs.
Clarity:
It is OK.
Significance:
It is an interesting idea to implement concurrent the generative model learning and the model inference.


**Time Spent Reviewing:**

3

---

> ### Author Response · Authors · 2021-08-09
> **Review Response (YoDe)**
>
> Q. what is the biological meaning of D_r in equation 6 ? D_r seems to be a recurrent connection matrix in the latent layer, but it is diagonal and does not play a role when k_t=1.
>
> A. As a diagonal recurrent matrix, D_r corresponds to a self-coupling parameter (which we think of as an intrinsic neural process rather than an autapse). Values of D_r close to 1 cause the neuron to continue firing at the same rate, and values of D_r close to 0 cause it to dampen its firing activity from time step to time step: therefore D_r is roughly interpretable as the resistance for a leak current in the apical dendritic compartment; the neuron dynamics could be reparameterized to make this correspondence closer.
> It is important to note that k_t = 1 only if the network is in its ‘inference’ phase, where activity is dominated by basal, rather than apical inputs. That means that k_t does not affect the dynamics of the neuron, but does affect the dynamics of learning.
>
> Q. the neuron should receive both top-down (apical) and bottom-up (basal) inputs concurrently，the variable λ_t induced by dopamine may control the learning phase (the infer phase and generative phase), but should not also control the apical inputs and basal inputs.
>
> A. To clarify, in our current derivation, lambda_t determines both learning and neuron dynamics: during the inference phase (lambda_t = 1), basal inputs determine neural activity, and apical synapses learn; conversely, during the generative phase (lambda_t = 0), apical inputs determine neural activity, and basal synapses learn. We have not hypothesized that dopamine controls or instantiates the lambda_t signal in the paper; rather, we suggest that inhibitory interneurons would provide this function, which have been shown to control apical and basal interactions as well as synaptic plasticity in the hippocampus (Leão et. al, 2012; Guerreiro et. al, 2020; Saudargiene & Graham 2015), though more evidence is needed to extend these results to sensory cortices. As our derivation currently stands, lambda_t must gate both neural activity and learning. However, as noted in the Discussion, extending the algorithm to situations where both basal and apical inputs to neurons influence neural dynamics at the same time (without oscillating lambda_t) may be possible.

---

### Official Review · Reviewer_ZFwx · 2021-07-17

**Rating:** 7
**Confidence:** 4

**Summary:**

This work combines probabilistic learning in the tradition of the Helmholtz machine with the recent hypothetical ideas about apical dendrites carrying top-down learning signals (expectation) to implement BP using local learning rule promoted earlier by numerous authors such as Larkum,  Lillicrap and Richards.  There are innovations. The impression learning mechanism proposed does sleep-wake in real-time.  It avoids the offline sleep phase in learning by using a global switching signal that can briefly substitute real incoming data with generative samples to enable learning continuously in a way that minimally perturbs natural neural trajectories.  This presents another new approach that allows the brain to learn generative models through local synaptic plasticity while concurrently performing sampling-based approximate inference wrt these models.  Incidentally, this work has been presented in COSYNE 2021 by the authors.



**Ethical Concerns:**

none.

**Limitations And Societal Impact:**

Broader impact section is missing.

**Main Review:**

Conceptually, the advance might be rather incremental.   It synthesizes many current existing ideas into a coherent framework.  It is well-written,  with mathematical rigor (more so than most of the earlier models). One weakness is that the approach has not been tested rigorously and compared against BP and other biologically-plausible BPs on benchmark datasets ( MINST, CIFAR or ImageNet) to evaluate its performance or to establish its competency as are customarily done nowadays in similar submissions.  Besides, as the authors admit, there is no biological evidence for this global switching signal.  While its relevance to understanding the brain is rather limited at present, it might stimulate new experimental research.

**Time Spent Reviewing:**

3 hours.

---

> ### Author Response · Authors · 2021-08-09
> **Review Response (ZFwx)**
>
> Q. One weakness is that the approach has not been tested rigorously and compared against BP and other biologically-plausible BPs on benchmark datasets (MINST, CIFAR or ImageNet) to evaluate its performance or to establish its competency as are customarily done nowadays in similar submissions.
>
> A. Impression learning is designed to handle temporal sequences: this constitutes one of its primary improvements over many other biologically plausible learning algorithms, which are designed for static data. When operating on single static images, impression learning is equivalent to the Wake/Sleep algorithm-- a point that we should (and will) make clearer in the text. Because of this, we focused the paper on sensory stimuli with temporal autocorrelations.
> We chose to use the Free Spoken Digits Dataset, because it is very simple (analogous to MNIST in the audio domain), but requires a generative model to capture the high-dimensional, naturalistic temporal relationships between data points. Within this context, we did compare impression learning’s performance to a version of neural variational inference (NVI~) and backpropagation through time (BPTT) (Fig. 2c-d; Appendices B and C); we will also include a more detailed comparison to the Wake/Sleep algorithm.
>
> Q. Besides, as the authors admit, there is no biological evidence for this global switching signal.
>
> A. We agree that additional experimental evidence will be necessary to fully validate our hypothesis, but it is worth noting some current supporting evidence. In particular, inhibitory interneurons have been shown to control apical and basal interactions as well as synaptic plasticity in the hippocampus (Leão et. al, 2012; Guerreiro et. al, 2020; Saudargiene & Graham 2015), and so may offer a plausible mechanism for the switching in our model. Whether such a mechanism could be truly ‘global,’ and whether these results extend to early sensory cortices remains unclear.
>
> Q. Broader impact section is missing.
>
> A. As we understood, an explicit broader impact section was not required this year, but we will add to our Discussion a note that we do not foresee any substantial negative societal impacts (as per the checklist).

---

> > ### Comment · Reviewer_ZFwx · 2021-08-27
> > **Post-rebuttal comment.**
> >
> > OK. Thanks for the explanation.  I will maintain my "good paper -- accept" score.

---

### Decision · Program_Chairs · 2021-09-27

**Decision:**

Accept (Poster)

**Comment:**

This paper introduces an unsupervised local plasticity rule to perform approximate online Bayesian inference of latent structure from sensory stimuli. All reviewers agree on the novelty and significance of the paper. However, reviewers brought up concerns about biological plausibility of some elements of the proposed algorithm, which should be addressed.